# SpecTRA: Spectral Transformer for Graph Representation Learning

## Abstract

Several positional and structural encodings are recently proposed into vanilla transformer architecture to overcome its inability to model graph's positional invariance and topology. However, in addition to graph topology, graph signals could be multi-channeled and contain heterogeneous information which cannot be inherently captured by the transformer. Hence, we propose an approach to induce a spectral module into the transformer architecture to enable decomposition of graph spectrum and selectively learn useful information akin to filtering in the frequency domain. Empirical results suggest that SpecTRA provides homogeneous performance gain against vanilla transformer across all tasks on standard benchmarks. Furthermore, incorporating SpecTRA instead of vanilla transformer model with recently proposed position encoding schemes have resulted in comparable or better performance than existing transformer and GNN based architectures.

## 1 Introduction

Several graph neural network (GNN) approaches have been devised as generic and efficient framework to learn from graph-structured data for tasks such as graph classification, node classification, graph regression, and link property prediction (Zhou et al., 2020). Among them, message-passing GNNs (MPGNNs) have been prominently used to obtain latent encoding of graph structures, achieving good results on related tasks (Gilmer et al., 2017; Veličković et al., 2018; Xu et al., 2019a). Although effective, these methods suffer from performance issues such as over-smoothing (Zhao & Akoglu, 2020), suspended animation (Zhang & Meng, 2019), and over-squashing (Alon & Yahav, 2021). Recently, researchers (Zhang et al., 2020; Li et al., 2018) have attempted to use transformers (Vaswani et al., 2017) for graph representation learning. However, transformers are inherently incapable of learning the topological information of graphs (Dwivedi et al., 2020). Hence, more recent works have added a gamut of effective positional and structural encoding methods (Mialon et al., 2021; Kreuzer et al., 2021) to alleviate limitation of transformers to learn topological information.

**Background and Hypothesis:** Viewed from classical signal processing domain, signals on a graph could contain heterogeneous information spread over a wider frequency domain (Ortega et al., 2018). The message passing graph convolutional networks (MPGCNs) such as GCN (Kipf & Welling, 2017) and GAT (Veličković et al., 2018) have low-pass characteristics (Muhammet et al., 2020). Furthermore, researchers aim go beyond static low or highpass filters to design adaptive filters to capture entire graph spectrum in GNNs (Gao et al., 2021; Bo et al., 2021; Chien et al., 2021). We argue that a similar issue plagues the transformers that cannot effectively segregate the noise from the signal for the entire spectral components of a graph. Gleaning from the work on MPGNNs as transformer can be special case of GNNs (Joshi, 2020), we hypothesize that inducing transformers with the ability to selectively use the signals in the frequency domain will enable better representation learning on a broader range of tasks. As graphs have different topologies, we further postulate that having an approach to dynamically select filters, as opposed to dataset-specific single static filter, will be more expansive to permit effective learning of graph representations using transformers.

**Proposed Approach:** This work, instead of devising a new position encoding scheme, proposes a novel direction. Our work aims to empower transformers to learn the essential components of the graph spectrum while filtering out the noise. We propose SpecTRA that effectively integrates the attention of the transformer with the spectrum of the graph. SpecTRA consists of a filtering module built on top of the vanilla transformer (Vaswani et al., 2017) to learn distinct *filter coefficients* for

each input graph. The transformer naturally gives diverse *attention sub-spaces*, which are utilized by SpecTRA to design multiple filters covering a broad spectrum of the graph. Specifically, we utilize the spatial information learned by the attention heads of the transformer to dynamically decide the *filter coefficients*. This enables interpreting the association between a particular frequency component with each head for a given graph. Furthermore, it also helps select the useful range of information in the frequency domain for different sub-spaces (attention heads in SpecTRA's case). Our rationale emerges from the following: 1) If the graph spectrum consists of several underlying components, which are relevant for different sub-graphs, then it is paramount to dynamically learn *filter coefficient*. This also provides an extra advantage concerning *interpretability* in spectral space per graph (cf., section 5.1). 2) In cases where certain classes are skewed, a task-specific filter would learn generic coefficients for the majority class, keeping a limit on the number of filters. SpecTRA handles these special cases by learning filters after observing the input graph (cf., section A.5).

**Contributions:** To summarize, this work make the following key contributions:

- Our primary contribution is SpecTRA, a novel approach for empowering vanilla transformer with the ability to perform *graph-specific* filtering in multiple sub-spaces of the attention heads. We study the efficacy of SpecTRA by conducting extensive experiments on standard benchmark datasets of graph classification/regression and node classification resulting in superior performance compared to vanilla transformer.

- We provide an exhaustive empirical study on the effect of inducing a variety of recently proposed transformer-based positional encoding schemes (Dwivedi et al., 2020; Kreuzer et al., 2021; Mialon et al., 2021) into the SpecTRA architecture. Empirical results show comparable or better performance against current state-of-the-art transformer and GNN based approaches. Hence, transformers, when induced with the ability to decompose and attend to signals spectrally, can effectively capture representations on a wide range of tasks complementing the effect of position encoding schemes.

The remainder of the paper is organized as follows: we describe the related works in section 2. Section 3 provides the preliminaries and problem definition. In section 4, we explain the proposed approach for graph specific dynamic filtering. Dataset details, experimental results, and ablations are given in sections 5 and 5.1. Section 6 concludes the paper.

## 2 RELATED WORK

**GNNs and Graph Transformers:** In this section, we stick to the work closely related to our approach (detailed survey in (Chen et al., 2020b)). Since the early attempts for GNNs (Scarselli et al., 2008), many variants of the message passing scheme were developed for graph structures such as GCN (Kipf & Welling, 2017), GIN (Xu et al., 2018), and GraphSAGE (Hamilton et al., 2017). The message passing paradigm employs neural networks for updating representation of neighboring nodes by exchanging messages between them. The use of transformer-style attention to GNNs for aggregating local information within the graphs is also an extensive research topic in recent literature (Thekumparampil et al., 2018; Shi et al., 2020; Li et al., 2018). Dwivedi & Bresson (2020) use the eigenvectors of the graph laplacian to induce positional information into the graph. Kreuzer et al. (2021) propose a learnable position encoding module that applies a transformer on the eigenvectors and eigenvalues of the graph laplacian. Ying et al. (2021) provide the concept of relative positional encoding in which the positional information is induced in the attention weights rather than in the input by obtaining correlation matrices of the spatial, edge, and centrality encoding. Similarly, Mialon et al. (2021) induce relative position information in the form of diffusion and random walk kernels along with structural information using the Graph Convolutional Kernel network (GCKN). In contrast with existing transformer-based attempts that inherit vanilla transformer model (Vaswani et al., 2017) for inducing position and topological encoding, we aim to improve the capabilities of the vanilla transformer model by empowering it to decompose the spectrum of a graph.

**Filters on Graphs:** Filtering in the frequency domain is generalized to graphs using the spectral graph theory (Chung et al., 1997; Shuman et al., 2013). The GCN model (Kipf & Welling, 2017) and variants such as (Zhang & Meng, 2019) approximate convolution for graph structures in the spatial domain. However, these models suffer from operating in the low-frequency regime, leaving rich information in graph data available in the middle- and high-frequency components (Gao et al.,

2021). Other approaches in the spectral domain attempt to reduce the computationally complex eigen decomposition of the laplacian by adopting certain functions of the graph laplacian such as Chebyshev polynomials (Defferrard et al., 2016), Cayley polynomials (Levie et al., 2018), and auto regressive moving average (ARMA) filters (Isufi et al., 2016). These approaches focus on designing specific filters with desirable characteristics such as bandpass and highpass. Our work focuses on the design of learnable filters, whose frequency response can be represented in polynomial functions in multiple sub-spaces of the signal. Gao et al. (2021) is closely related to part of our work which also designs filter banks, albeit task-specific, for heterogeneous and multi-channel signals on graphs. However, our work differs from Gao et al. (2021) in that we learn *graph-specific* filters to enable *interpretability* in spectral space per graph and filtering into the transformer architecture.

## 3 PRELIMINARIES AND PROBLEM DEFINITION

**Graph Fourier Transform:** We denote a graph as $(\mathcal{V}, \epsilon)$ where $\mathcal{V}$ is the set of $N$ nodes and $\epsilon$ represents the edges between them. The adjacency matrix is denoted by $A$. Here, we consider the setting of an undirected graph, hence, $A$ is symmetric. The diagonal degree matrix $D$ is defined as $(D)_{ii} = \sum_j (A)_{ij}$. The normalized laplacian $L$ of the graph is defined as $L = I - D^{-\frac{1}{2}} A D^{-\frac{1}{2}}$. The laplacian $L$ can be decomposed into its eigenvectors and eigenvalues as:

$$L = U \Lambda U^*$$

where U is the $N \times N$ matrix; the columns of which are the eigenvectors corresponding to the eigenvalues $\lambda_1, \lambda_2, \ldots, \lambda_N$ and $\Lambda = \text{diag}(\lambda_1, \lambda_2, \ldots, \lambda_n])$. Let $X \in R^{N \times d}$ be the signal on the nodes of the graph. The Fourier Transform $\hat{X}$ of $X$ is then given as: $\hat{X} = U^* X$. Similarly, the inverse Fourier Transform is defined as: $X = U\hat{X}$. Note $U^*$ is the transposed conjugate of $U$. By the convolution theorem (Blackledge, 2005), the convolution of the signal $X$ with a filter G having its frequency response as $\hat{G}$ is given by (below, $v_m$ is the $m^{th}$ node in the graph):

$$(X * G)(v_m) = \sum_{k=1}^{n} \hat{X}(\lambda_k)\hat{G}(\lambda_k)U(v_m) = \sum_{k=1}^{n} (U^* X)(\lambda_k)\hat{G}(\lambda_k)U(v_m, \lambda_k) = U\hat{G}(\Lambda)U^* X(v_m) \quad (1)$$

**Transformers and Position encoding:** The recent works (Dwivedi & Bresson, 2020; Kreuzer et al., 2021) established the necessity of encoding schemes for inducing positional encoding into the transformer architecture. The relative encoding schemes proposed by GraphiT (Mialon et al., 2021) use diffusion and random walk kernels for relative positional encodings. The diffusion kernel $K_D$ for a graph with laplacian $L$ is given by the equation below,

$$K_D = e^{-\beta L} = \lim_{p \to \inf} (I - \frac{\beta}{p}L)^p \quad (2)$$

For physical interpretation, diffusion kernels can be interpreted as the amount of a substance that accumulates at a given node if injected into another node and allowed to diffuse through the graph. Similarly, the random walk kernel generalizes this notion of diffusion for fixed-step walks on the graph. It is described by the equation below,

$$K_{pRW} = (I - \gamma L)^p \quad (3)$$

We can see that above equation becomes the diffusion kernel if $\gamma = \frac{\beta}{p}$ and $p \to \inf$. However, the difference with respect to the diffusion kernel is that the random walk kernel is sparse. GraphiT also makes use of GCKN (Chen et al., 2020a) for learning structural position encoding and we inherit it in our approach describe in Section 4. We refer the readers to (Mialon et al., 2021) for more details.

**Spectral GNN:** Spectral GNNs rely on the spectral graph theory (Chung et al., 1997). Consider a graph with $U$ as its eigen vectors, $\lambda$ the eigen values, and $L$ the laplacian. The graph convolution operation in the frequency domain can be written as below,

$$H_j^{l+1} = \sum_{i=1}^{d_l} U diag(G_{i,j,l}) U^* H_i^l \quad (4)$$

where $H_j^l$ is the feature vector of the $j^{th}$ node in the $(l)^{th}$ layer, $d_l$ is the dimension of the signal in the $l^{th}$ layer, $G_{i,j,l}$ is the learnable weight vector for the $i^{th}$ and $j^{th}$ node in the $l^{th}$ layer. This

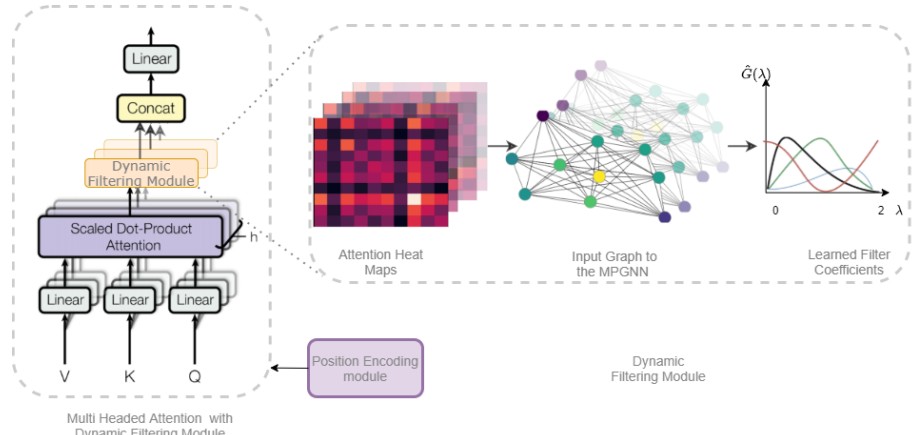

Figure 1: SpecTRA architecture. Besides introducing a dynamic filtering module for learning multiple filters per attention head, position encoding schemes are used in a plug-and-play manner.

formulation is non-transferable to multigraph learning problem (Muhammet et al., 2020). Thus, $G_{i,j,l}$ is re-parametrized as below,

$$G_{i,j,l} = B[W_{i,j}^{l,1}, W_{i,j}^{l,2}, \dots W_{i,j}^{l,S}]$$

where $B \in R^{N \times S}$, $S$ is the number of convolutional supports and $W^{l,s}$ is the learnable matrix. The Equation 4 depends on the computation of the eigen vectors $U$ of $L$, which is computationally costly for large graphs. In our work, we consider the polynomial approximations as proposed by Hammond et al. (2011). Specifically, the frequency response of the desired filter can be approximated as,

$$\hat{G} = \sum_{k=0}^{K} \alpha^k T_k(\tilde{\Lambda}) \tag{5}$$

where $T(k)$ is the polynomial basis such as Chebyshev polynomials (Defferrard et al., 2016), $\tilde{\Lambda} = \frac{2\Lambda}{\lambda_{max}} - I$, $\lambda_{max}$ is the maximum eigen value and $\alpha^k$ is the corresponding *filter coefficients*. A recursive formulation could be used for the Chebyshev polynomials with basis $T_0(x) = 1$, $T_1(x) = x$ and beyond that $T_k(x) = 2xT_{k-1}(x) - T_{k-2}(x)$. Thus, the convolution operation in Equation 1 can be approximated as

$$X * G \approx U(\sum_{k=0}^{K} \alpha^k T_k(\tilde{\Lambda}))U^* X = \sum_{k=0}^{K} \alpha^k T_k(U\tilde{\Lambda}U^*)X = \sum_{k=0}^{K} \alpha^k T_k(\tilde{L})X \tag{6}$$

This corresponds to an FIR filter of order $K$ (Smith et al., 1997). Setting $\alpha$ as a learnable parameter help us learn the filter for the upstream task.

**Problem definition** In this work, we aim to learn the *filter coefficients* from the attention weights of the transformer. Formally, given the transformer attention for the head $h$ at layer $l$ as $A^{h,l} \in R^{N \times N}$, we aim to define a mapping $M : R^{N \times N} \to R^K$ where $K$ is the filter order. The mapping $M$ would take us from the space in the attention weights to the filter coefficient space.

## 4 SPECTRA:SPETRAL TRANSFORMER

In this section, we present our approach, SpecTRA. We first describe the notion of dynamic filtering module of SpecTRA architecture followed by its position encoding module (cf., Figure 1).

### 4.1 MULTI HEADED ATTENTION WITH DYNAMIC FILTERING MODULE

We view the graph as set of node features to be fed to the vanilla transformer (Vaswani et al., 2017) that learns the pairwise similarity between these nodes using its attention mechanism as follows:

$$AttentionWeights^h(Q, K) = softmax(\frac{QK^T}{\sqrt{d_{out}}}) \tag{7}$$

Here, $Q^T = W_Q^h X^T$ and $K^T = W_K^h X^T$ where $W_Q^h, W_K^h \in R^{d_{out} \times d_{in}}$ are the projection matrices for the query and key respectively for the head $h$. The output $X^h$ at the head $h$ can be obtained from:

$$X^h = Attention^h(Q, K, V) = softmax(\frac{QK^T}{\sqrt{d_{out}}})V \tag{8}$$

**Learning filter coefficients**: In order to obtain the *filter coefficients* from the weights of each attention head, we use a message passing framework. The spectral and spatial domains are interchangeable, and transformers traditionally learn good latent representations of spatial domain (Vaswani et al., 2017; Devlin et al., 2019). Hence, attention map provides inference to the connectivity of a graph implicitly, using which, we can deduce the frequency band to filter out the noise. If each node's signal $x_i$ is considered with its neighborhood $\mathcal{N}(x_i)$ which represents the non-zero attention weights obtained from Equation 7, then the message passing is defined as:

$$x_{im}^l = A(x_j^l | x_j \in \mathcal{N}(x_i)), x_i^{l+1} = U(x_i^l, x_{im}^l)$$

where $x_i^l$ and $x_{im}^l$ are the signals and the aggregated message at node $x_i$ in the layer $l$. Here, $A$ and $U$ are the aggregation and update functions respectively. This framework enables the usage of popular message passing schemes such as GCN (Kipf & Welling, 2017). For example, in GCN, the aggregation is a simple projection of the signals followed by a summation weighted by the normalized laplacian with self loops. Next, the updation is an activation function such as ReLU. The final equation is now represented as below:

$$x_i^{h,l+1} = \sigma(\sum_{x_j \in \mathcal{N}(x_i)} L[i,j](x_j^{h,l})^T W_p^l \quad) \tag{9}$$

where $x_j^{h,l}$ is the node embedding for the $j^{th}$ node in the $l$-th layer of the GNN(to learn coefficients) for the $h$-th head of the transformer, $L = I - D^{-\frac{1}{2}} A D^{-\frac{1}{2}}$ is the normalized laplacian and $W_p^l$ is the learnable projection matrix at layer $l$. This GNN is common for all the heads and at all layers of the transformer. Here, the node embeddings reside in the coefficient space i.e. $X \in R^{N \times K}$, where $K$ is the order of the filter. For learning the *filter coefficients* we initialize the node embeddings with a prior of the filter depending on the task. For an all-pass filter we could use a vector of all ones as the initialization. This choice of prior is justified by property 4.1 (cf., Appendix A.1 for proof).

**Property 4.1.** *The filter coefficients consisting of the vector of all ones is an all-pass filter.*

After the $L$-th message passing, the vector obtained from Equation 9 is given to a readout function such as global average pooling followed by a simple feed forward network to obtain the final *filter coefficients* $\alpha^h$, per attention head, as below:

$$\alpha^h = MLP(\frac{1}{N}\sum_{i=1}^{N} x_i^{h,L}) \tag{10}$$

where $x_i^h$ is the vector at node $x_i$ in the $h$-th attention head obtained from Equation 9.

**Dynamic Filtering Module**: We use the filter coefficient $\alpha^h$ to obtain the appropriate filter frequency response defined as:

$$\hat{G}^h = \sum_{k=0}^{K} \alpha^h[k]T_k(\tilde{L}) \tag{11}$$

where K is the filter order. The desired filter response $H^h$ at head $h$ can then be obtained from $\hat{G}^h$ as observed in Eq 6 as:

$$H^h = X^h * G^h = U\hat{G}^h(\tilde{\Lambda})U^* X^h = \hat{G}^h(\tilde{L})X^h = \sum_{k=0}^{K} \alpha^h[k]T_k(\tilde{L})X^h$$

The filter outputs from each head is concatenated to get the filtered output which is further concatenated with the attention output $X$, followed by an MLP, with appropriate normalizations, for the output of the encoder layer:

$$H = \|_h H^h, \quad X = MLP(\|_h X^h), \quad X^a = Norm(MLP(X \oplus H))$$

This could then be used in the upstream task of classification, regression, etc. In order to learn distinct *filter coefficients* for each head we add a regularization term to the objective. The regularization tries to keep the coefficient vectors orthogonal to each other. It does so by taking the Frobenius norm (Horn & Johnson, 1990) of the gram matrix of $X$ whose columns consist of the coefficient vector of each head. Formally, define $\alpha^i \in R^k$ and $X = [\alpha^1, \alpha^2, \dots \alpha^h] \in R^{k \times h}$ where $h$ is the number of heads and $k$ is the filter order. The regularization term is given by $\|(X^T X) \odot (\mathbf{1} - I)\|_2$, where $I \in R^{h \times h}$ is the identity matrix and $\odot$ is the hadamard product. The below theorem (proof in Appendix A.1) justifies the proposed method.

**Theorem 4.1.** *Assume the desired filter response $G(x)$ has $m + 1$ continuous derivatives on the domain $[-1, 1]$. Let $S_n^T G(x)$ denote the $n^{th}$ order approximation by the polynomial(chebyshev) filter and $S_n^{T'} G(x)$ denote the learned filter, $C_f$ be the first absolute moment of the distribution of the fourier magnitudes of $f$ (function learned by the network), $h$ the number of hidden units in the network and $N$ the number of training samples. Then the error between the learned and desired frequency response is bounded by the below expression*

$$|G(x) - S_n^{T'} G(x)| = \mathcal{O}\left(\frac{nC_f^2}{h} + \frac{hn^2}{N} log(N) + n^{-m}\right)$$

The above theorem argues that the Chebyshev polynomials are able to learn any smooth function. Using injective aggregators in GNNs and universal approximators in MLP, we can approximate these coefficients to the desired precision given by the bounds. The input to the GNNs could be divided into two parts: the input graph indicating the connectivity of nodes and the signals on the graph. In this work, we aim to study if the spectral components could be identified by using the *spatial connectivity pattern* of the signals on the original graph obtained from the attention maps of the transformer. This justifies the proposed architecture, in which a graph is constructed with the edge weights taken from the attention weights and fed to a GNN for learning the filter coefficients. These coefficients are then used in a spectral GNN in which the input is the original graph with node embeddings as learned from the transformer. The output of this is then fed to further transformer layers, and the process is repeated. The input to the first transformer is the node/edge attributes itself. We leave using the connectivity structure imposed by the original graph for future works.

## 4.2 POSITIONAL ENCODING SCHEMES

In SpecTRA, we learn the pairwise similarity between graph nodes inheriting the attention mechanism of vanilla transformer as follows:

$$Attention^h(Q, K, V) = softmax\left(\frac{QK^T}{\sqrt{d_{out}}}\right)V \tag{12}$$

Here $Q^T = W_Q^h X^T$, $K^T = W_K^h X^T$ and $V^T = W_V^h X^T$, where $W_Q^h, W_K^h, W_V^h \in R^{d_{out} \times d_{in}}$ are the projection matrices for the query, key and values respectively for the head $h$. Following GraphiT (Mialon et al., 2021), we share the weight matrices for the query and key matrices for learning a positive semi-definite kernel. For the input features we use the node attributes, if provided, along with the static laplacian position encoding, as in (Dwivedi & Bresson, 2020). The static encoding in the form of the laplacian eigen vectors is simply added to the node embeddings. For the relative positional encoding we follow GraphiT and use the diffusion ($K_D$) and random walk ($K_{rw}$) kernels (cf., Equations 2 and 3). The attention using the relative positional encoding schemes is now:

$$Attention(Q, V) = softmax\left(\exp\left(\frac{QQ^T}{\sqrt{d_{out}}}\right) \cdot K_p\right)V \tag{13}$$

Here, $K_p$ is the respective kernel being used i.e. $K_p \in K_D, K_{rw}$. We also borrow from the positional encoding scheme of SAN (Kreuzer et al., 2021) that allow usage of the edge features $E \in R^{N \times N \times d_{in}}$. Formally, the attention weights $w_{ij}^{kl}$ between the nodes $i$ and $j$ in the $l^{th}$ layer and

$k^{th}$ attention head is given by the below equations

$$
\hat{w}_{ij}^{kl} = \begin{cases} \dfrac{W_Q^{1,k,l}X[i]^T \odot W_K^{1,k,l}X[j]^T \odot W_E^{1,k,l}E[i,j]^T}{d_{out}}, & \text{if } i \text{ and } j \text{ are connected in sparse graph} \\[2ex] \dfrac{W_Q^{2,k,l}X[i]^T \odot W_K^{2,k,l}X[j]^T \odot W_E^{2,k,l}E[i,j]^T}{d_{out}}, & \text{otherwise} \end{cases}
$$

$$
w_{ij}^{kl} = \begin{cases} \dfrac{1}{1+\gamma} softmax(\sum_{d_k} \hat{w}_{ij}^{kl}), & \text{if } i \text{ and } j \text{ are connected in sparse graph} \\[2ex] \dfrac{\gamma}{1+\gamma} softmax(\sum_{d_k} \hat{w}_{ij}^{kl}), & \text{otherwise} \end{cases}
$$

where $W_Q^{1,k,l}, W_K^{1,k,l}, W_E^{1,k,l}$ are the projection matrices corresponding to the query, key and edge vectors for the real edges and $W_Q^{2,k,l}, W_K^{2,k,l}, W_E^{2,k,l}$ are the projection matrices corresponding to the respective vectors for the added edges in the $k^{th}$ head and $l^{th}$ layer as in (Kreuzer et al., 2021). We further employ GCKN (Chen et al., 2020a) to encode graph's topological properties.

**Limitations** The space and time complexity of our method is $\mathcal{O}(N^2)$ for full attention. This could be alleviated by using sparse attention (cf., Table 2). One may also use kernel methods as in (Choromanski et al., 2021) to reduce the number of nodes in the graph and then apply the transformer on the reduced graph. Similar to vanilla transformer, SpecTRA cannot induce positional encoding on its own. It is yet to be explored if learning position encodings could be incorporated into the proposed method. We leave these directions of exploration to future works.

## 5 EXPERIMENT RESULTS

We aim to answer following research questions: **RQ1:** Can multi-headed attention combined with *dynamic graph filtering* improve SpecTRA's ability over base transformer for graph representation learning? **RQ2:** What is the impact of recently proposed *position encoding* schemes on SpecTRA? **RQ3:** What is the efficacy of *graph-specific* filters on the performance of SpecTRA?

**Datasets, Settings and Baselines**: We use widely popular datasets (Kreuzer et al., 2021; Mialon et al., 2021; Hu et al., 2020): MUTAG, NCI1, and the OGBG-MolHIV for graph classification; PATTERN and CLUSTER for node classification; and ZINC for graph regression task. Further details are in the Appendix A.2. We borrow experiment settings from (Mialon et al., 2021; Kreuzer et al., 2021). The GNN and transformer baselines are listed in Table 5.1.

**SpecTRA Configurations:** For a fair and exhaustive comparison, we provide six variants of Spec-TRA: 1) *SpecTra-Base* to compare against vanilla transformer without position encoding module (cf., section 4.2), 2) *SpecTRA+LapE* contains static position encoding from Mialon et al. (2021), 3) *SpecTRA+3RW* consists of position encoding based on 3-step RW kernel from Mialon et al. (2021), 4) *SpecTRA+GCKN+3RW* that uses GCKN (Chen et al., 2020a) in addition with 3-step RW kernel to induce graph topology, 5) *SpecTRA+LPE+Full* with learnable position encoding (Kreuzer et al., 2021) with full attention. 6) *SpecTRA+LPE+Sparse* inheriting learnable position encoding from Kreuzer et al. (2021), with sparse attention on the graphs.

### 5.1 RESULTS

SpecTRA-Base outperforms vanilla transformer (Table 1) across all datasets establishing the positive impact of combining *dynamic filtering* with multi-headed attention into SpecTRA architecture (successfully answering **RQ1**). We observe (Table 2) that for the smaller MUTAG and NCI1 datasets, SpecTRA achieves the best results against baselines using the structural encoding of GCKN. It indicates that these datasets benefit more from structural information. The results of the other SpecTRA variants are comparable or better than the transformer-based (SAN, GraphiT-LapE, GraphiT-3RW) and GNN baselines. On OGBG-MolHIV and PATTERN/CLUSTER's graph and node classification tasks, learnable position encoding has most positive impact on SpecTRA. Graphformer reports the highest value on MolHIV. However, its parameters are 47M compared to $\approx 500$K from Spec-TRA+LPE, GCN-based models, and SAN. Notably, we see that SpecTRA performs exceptionally well on the graph regression task of ZINC with a relative decrease in the error of up to $50\%$. Potential reason could be SpecTRA's ability to learn diverse filters that are distinct for each graph (c.f,

| Models | MUTAG | NCI1 | ZINC | MolHIV | PATTERN | CLUSTER |
|---|---|---|---|---|---|---|
| Vanilla Transformer | 82.2 ± 6.3 | 70.0 ± 4.5 | 0.696 ± 0.007 | 65.22 ± 5.52 | 75.77 ± 0.4875 | 21.001 ± 1.013 |
| SpecTRA-Base (ours) | 87.2 ± 2.6 | 73.7 ± 1.4 | 0.412 ± 0.004 | 65.77 ± 3.838 | 78.65 ± 2.509 | 30.351 ± 2.669 |

Table 1: Results on graph/node classification/regression Tasks (**RQ1**)). Higher (in green) value is better (except for ZINC). Means and uncertainties are derived from four runs.

| Models | MUTAG % ACC | NCI1 % ACC | ZINC MAE | MolHIV % ROC-AUC | PATTERN % ACC | CLUSTER % ACC |
|---|---|---|---|---|---|---|
| GCN (Kipf & Welling, 2017) | 78.9±10.1 | 75.9 ± 1.6 | 0.367 ± 0.011 | 76.06 ± 0.97 | 71.892 ± 0.334 | 68.498 ± 0.976 |
| GatedGCN (Bresson & Laurent, 2017) | - | - | 0.282 ± 0.015 | - | 85.568 ± 0.088 | 73.840 ± 0.326 |
| GraphSAGE (Hamilton et al., 2017) | - | - | 0.398 ± 0.002 | - | 50.492 ± 0.001 | 63.844 ± 0.110 |
| GAT (Veličković et al., 2018) | 80.3 ± 8.5 | 74.8 ± 4.1 | 0.384 ± 0.007 | - | 78.271 ± 0.186 | 70.587 ± 0.447 |
| PNA(Corso et al., 2020) | - | - | 0.142 ± 0.010 | 79.05 ± 1.32 | - | - |
| GIN (Xu et al., 2018) | 82.6 ± 6.2 | 81.7 ± 1.7 | 0.526 ± 0.051 | 75.58 ± 1.40 | 85.387 ± 0.136 | 64.716 ± 1.553 |
| Graphormer (Ying et al., 2021) | - | - | 0.122 ± 0.006 | 80.51 ± 0.53 | - | - |
| GT-sparse(Dwivedi & Bresson, 2020) | - | - | 0.226 ± 0.014 | - | 84.808 ± 0.068 | 73.169 ± 0.662 |
| GT-full (Dwivedi & Bresson, 2020) | - | - | 0.598 ± 0.049 | - | 56.482 ± 3.549 | 27.121 ± 8.471 |
| SAN-Sparse(Kreuzer et al., 2021) | 74.1 ± 2.6* | 80.5 ± 1.3* | 0.198 ± 0.004 | 76.61 ± 0.62 | 81.329 ± 2.150 | 75.738 ± 0.106 |
| SAN-full(Kreuzer et al., 2021) | 71.9 ± 2.9* | 75.1 ± 1.5* | 0.139 ± 0.006 | 77.85 ± 0.65 | 86.581 ± 0.037 | 76.691 ± 0.247 |
| GraphiT-LapE (Mialon et al., 2021) | 85.8 ± 5.9 | 74.6 ± 1.9 | 0.507 ± 0.003 | 65.10 ± 1.76* | 76.701 ± 0.738* | 18.136 ± 1.997* |
| GraphiT-3RW (Mialon et al., 2021) | 83.3 ± 6.3 | 77.6 ± 3.6 | 0.244 ± 0.011 | 64.22 ± 4.94* | 76.694 ± 0.921* | 21.311 ± 0.478* |
| GraphiT-3RW+GCKN (Mialon et al., 2021) | 90.5 ± 7.0 | 81.4 ± 2.2 | 0.211 ± 0.010 | 53.77 ± 2.73* | 75.850 ± 0.192* | 69.658 ± 0.895* |
| SpecTRA + LapE (ours) | 87.4 ± 2.6 | 75.4 ± 2.6 | 0.077 ± 0.001 | 66.80 ± 2.18 | 78.808 ± 1.662 | 19.366 ± 3.818 |
| SpecTRA + 3RW (ours) | 87.0 ± 2.6 | 78.5 ± 1.3 | 0.104 ± 0.005 | 59.95 ± 3.91 | 77.285 ± 1.146 | 68.572 ± 2.164 |
| SpecTRA + GCKN+ 3RW (ours) | 92.9 ± 1.6 | 83.0 ± 0.5 | 0.068 ± 0.002 | 53.50 ± 5.89 | 77.86 ± 0.573 | 67.507 ± 2.856 |
| SpecTRA + LPE+Full (ours) | 72.2 ± 1.6 | 73.8 ± 0.8 | 0.1836 ± 0.002 | 76.88 ± 0.573 | 86.52 ± 0.013 | 76.750 ± 0.296 |
| SpecTRA + LPE+Sparse (ours) | 72.2 ± 3.5 | 81.0 ± 1.5 | 0.1581 ± 0.001 | 78.10 ± 0.303 | 86.30 ± 0.024 | 77.224 ± 0.111 |

Table 2: Impact of external position encoding schemes (**RQ2**). For a dataset, Green and Grey cells represent the highest and the second best result, respectively. The baselines values are from (Kreuzer et al., 2021; Mialon et al., 2021) and its additional values with * are calculated by us.

Appendix A.5). An important observation is that *no position encoding scheme provides homogeneous performance gain across all datasets* for SpecTRA and it supports our choice to use them as a *plug-in* (answering **RQ2**). Hence, it is still an open research question to propose a position encoding scheme that provides homogeneous performance gain across all tasks agnostic of the datasets.

| Models | MUTAG | NCI1 | ZINC | MolHIV | PATTERN | CLUSTER |
|---|---|---|---|---|---|---|
| SpecTRA-Base | 87.2 ± 2.6 | 73.7 ± 1.4 | 0.412 ± 0.004 | 65.77 ± 3.838 | 78.65 ± 2.509 | 30.351 ± 2.669 |
| SpecTRA-ARMA | 85.1 ±1.3 | 72.2 ± 2.1 | 0.355 ± 0.007 | 65.45 ± 4.237 | 76.93 ± 0.279 | 20.390 ± 2.859 |
| SpecTRA-Static | 83.3 ± 1.3 | 70.4 ± 3.8 | 0.470 ± 0.002 | 67.10 ± 3.647 | 76.03 ± 0.861 | 20.995 ± 0.005 |

Table 3: Comparing SpecTRA-Base against (**RQ3**): 1) SpecTRA-Static that uses static filter and 2) SpecTRA-ARMA which changes the filter in base configuration to ARMA.

**Ablation Studies** We created another SpecTRA configuration (SpecTRA-Static) where the filter is static per dataset based on attention heads. Our idea here is to understand the impact of combining multi-headed attentions with a *dynamic filter*. From Table 3, we clearly observe the empirical advantage of SpecTRA-Base. More importantly, *graph-specific* filters aim for *interpretability* in spectral space per graph. Similar to attention weights, learning *graph-specific* filters facilitates the understanding of which nodes interact in the spectral domain and which filter is helpful for the task. In some cases, it may benefit from learning representations by aggregating from connected neighbors, which a low pass filter would do. Whereas in other cases, the task would benefit learning from unconnected distant nodes, which may be graph dependent. For instance, in Figure 3, we see that for sparse graphs (graph (a) - (c)) the filter response has a prominent magnitude for the lower components of the spectrum along with some components in the middle regions of the spectrum. On the other hand, for relatively dense graphs, we see a relatively less prominent low frequency response and many heads learning to focus on the higher frequency components of the spectrum. In a sense, this enables aggregation of nodes that are distant in the graph and helps in providing *interpretation* as to which nodes interact in the *spectral domain*. Note that existing *graph-specific* attention mechanisms, such as GAT, learn only low pass filters (Muhammet et al., 2020) and cannot perform such an aggregation. Also, the spectral GNNs (cf., section 2) that learn filters for the entire dataset are

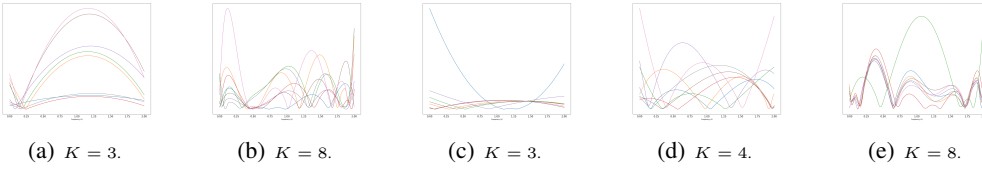

(a) $K = 3$.  (b) $K = 8$.  (c) $K = 3$.  (d) $K = 4$.  (e) $K = 8$.

Figure 2: Aggregate Filter Frequency response ($K$ is filter order). (a) $\sim$ (b) are on MUTAG and (c) $\sim$ (e) are on NCI1. X axis shows the normalized frequency with magnitudes on the Y axis.

unable to perform similar aggregation in a *graph-specific* manner which may be beneficial in some cases (cf., A.4). Also, the learned filters have different frequency responses specific to the dataset as they are tuned to the data characteristics (cf., Figure 2). These observations justify our rationale to propose *graph-specific* filters and successfully answering **RQ3**. We note, equation 11 represents

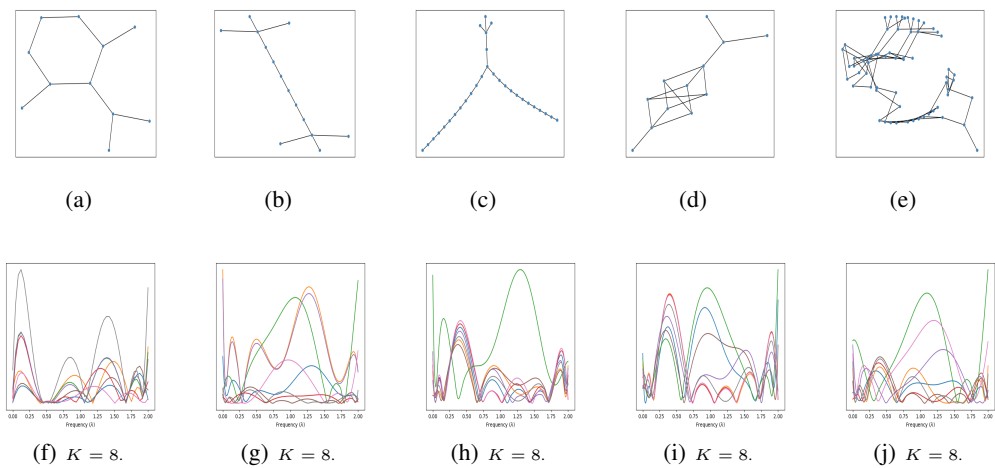

(a)   (b)   (c)   (d)   (e)

(f) $K = 8$.  (g) $K = 8$.  (h) $K = 8$.  (i) $K = 8$.  (j) $K = 8$.

Figure 3: Filter Frequency response on individual graphs. Graph (a) is from MUTAG and, (b) $\sim$ (e) are from the NCI1 dataset and Figures (f) $\sim$ (j) are the corresponding frequency responses. X axis shows the normalized frequency with magnitudes on the Y axis.

a polynomial filter using the Chebyshev polynomials (Defferrard et al., 2016). Polynomial filters (Hammond et al., 2011) are smooth and have restrictions that they cannot model filter responses with sharp edges. Hence, we devise SpecTRA-ARMA that uses rational filters such as ARMA (Isufi et al., 2016). Our idea here is to study the empirical efficacy of these filters. It can be observed from Table 3 that SpecTRA-ARMA resulted in lower performance than SpecTRA-Base on most of the datasets. However, on ZINC, there is an improvement using ARMA filters as compared to Chebyshev polynomial filters. Also on CLUSTER, we observer a sharp increase using SpecTRA-Base as compared to SpecTRA-Static but not with SpecTRA-ARMA. Hence, we conclude that filter response and empirical predictive performance could be orthogonal objectives, and it becomes a trade-off to decide which type of filter to apply for a given task.

## 6 CONCLUSION

In this work, we have introduced a novel transformer-based approach for graph representation learning. Our model SpecTRA aims to effectively learn multiple filters per attention head to capture heterogeneous information spread over a wider frequency domain. Furthermore, learning *graph-specific* filters provide *interpretability* in spectral space per graph, which is not the case in a dataset-specific static filter. Experiments on standard datasets of a variety of tasks suggest a clear empirical edge on vanilla transformer. Also, using recently proposed position encoding schemes as plug-ins in the flexible architecture of SpecTRA resulted in better or comparable state-of-the-art performance.

## 7 ETHICS AND REPRODUCIBILITY STATEMENT

In this work, we present significant progress in graph representation learning using transformers. Knowledge representation from the data (mainly graphs) is an important goal that human beings seek for reasoning along with the advancement of technology. Many recently proposed transformer approaches rely on static frequency filters (including one of our configuration SpecTRA-Static) compared to our proposed dynamic filtering approach. Furthermore, we employ widely used public datasets for graphs. When it comes to who may be disadvantaged from this research, we do not think it is applicable since our study of addressing the dynamic filtering capabilities of the transformer, which is still at an early stage. Additionally, there is no concrete evidence in the literature that signals on the graphs causes bias or ethical concerns for effectively learning graph representations.

Having said so, we are fully supporting the development of ethical and responsible AI. The potential bias in the standard public datasets that may lead to wrong knowledge needs to be cleaned or corrected with validation mechanisms. We are also aware of energy consumption of GPUs used in the empirical evaluations and support development of Green AI for sustainable development of ML research.

Regarding reproducibility: upon acceptance, we will publicly release code with detailed user instructions to reproduce our empirical values. We have provided an additional appendix section including GPU size and all other hyper parameters needed to reproduce this work. Also, we have made sure that in methodology, all relevant equations are presented in the main section of the paper to clarify our proposed approach.

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

# A APPENDIX

## A.1 THEORETICAL JUSTIFICATION

**Property A.1.** *The filter coefficients consisting of the vector of all ones is an all-pass filter*

*Proof.* Consider the filter response given by the chebyshev coefficients $T_n$ of the first kind as below

$$G(x) = \sum_{i=0}^{\infty} T_i(x) t^n$$

where $t^i$ represent the coefficients for the $i^{th}$ polynomial. It can be verified that the generating function for $G$ above could be given by the below equation

$$G(x) = \frac{1 - tx}{1 - 2tx + t^2} \tag{14}$$

setting $t = 1$ in this equation gives $G(x) = \frac{1}{2}$ which does not depend on $x$. Thus $G$ would behave as an all-pass filter. $\square$

Note that $t = 0$ is also an alternative but this would cause a degenerate learning of the filter coefficients which may always remain 0 after aggregation and the MLP layers (in the absence of bias).

**Theorem A.1.** *Assume the desired filter response $G(x)$ has $m + 1$ continuous derivatives on the domain $[-1, 1]$. Let $S_n^T G(x)$ denote the $n^{th}$ order approximation by the polynomial(chebyshev) filter and $S_n^{T'} G(x)$ denote the learned filter, $C_f$ be the first absolute moment of the distribution of the fourier magnitudes of $f$ (function learned by the network), $h$ the number of hidden units in the network and $N$ the number of training samples. Then the error between the learned and desired frequency response is bounded by the below expression*

$$|G(x) - S_n^{T'} G(x)| = \mathcal{O}(\frac{nC_f^2}{h} + \frac{hn^2}{N} log(N) + n^{-m})$$

*Proof.* We begin by defining a bounded linear functional $L$ on the space $C^{m+1}[-1, 1]$ with $m + 1$ continuous derivatives as below

$$LG = (S_n^T G)(x) - G(x)$$

Since $G(x)$ has $m + 1$ derivatives, it is approximated by a polynomial function of degree $n > m$. By Peano's kernel theorem (Peano, 1913), we can write $LG$ as below

$$(S_n^T G)(x) - G(x) = \int_{-1}^{1} G^{m+1}(t) K_n(x, t) dt \tag{15}$$

where

$$K_n(x, t) = \frac{1}{m!} S_n^T (x - t)_+^m - (x - t)_+^m \tag{16}$$

The notation $(. - t)_+^m$ indicates

$$(x - t)_+^m = \begin{cases} (x - t)^m & \text{if } x > t \\ 0 & \text{otherwise} \end{cases}$$

We note the $n^{th}$ order approximation of the chebyshev expansion of the function $Gx$ is given by

$$S_n^T G = \sum_{k=0}^n c_k T_k(x)$$

where,

$$c_k = \frac{2}{\pi} \int_{-1}^1 \frac{G(x)T_k(x)}{\sqrt{1-x^2}} dx$$

Thus we get $S_n^T (x-t)_+^m$ as

$$S_n^T (x-t)_+^m = \sum_{k=0}^n c_{km} T_k(x)$$

where,

$$c_{km} = \frac{2}{\pi} \int_t^1 \frac{(x-t)^m T_k(x)}{\sqrt{1-x^2}} dx$$

If a Graph Neural Network approximates $c_{km}$ in the proposed method, then we would have an error due to this approximation. We now find the bounds of this approximation. As the current method relies on the connectivity pattern due to the signals of the nodes of the graph, we desire a different graph for different signal patterns and thus different filter coefficients. Assuming an injective function (Xu et al., 2019b) for aggregation, we can always find a unique map for a unique graph corresponding to the frequency response. Now we need to learn a function that maps the output of the previous network to the coefficients of the desired frequency response. A classical neural network with $h$ hidden units could approximate this by the below bound (Barron, 1991)

$$\epsilon = c_{km}^{'} - c_{km} = \mathcal{O}(\frac{C_f^2}{h} + \frac{hd}{N} log(N))$$

where $d = n$ as the input dimension is the filter order, $N$ is the number of training samples. Intuitively the first term of the expression states that increasing the number of nodes $h$ decreases the error bounds and the second term acts as a regularizer to limit the increase in $h$ by the number of samples. The latter term can be thought of as enforcing the bound to obey the Occam's razor that parameters should not be increased beyond necessity.

$$C_f = \int |\omega||F| d\omega$$

$|F|$ is the Fourier magnitude distribution of $f$ which is the function to be learned by the network. Thus we get,

$$\epsilon = \mathcal{O}(\frac{C_f^2}{h} + \frac{hn}{N} log(N)) \tag{17}$$

We represent by $S_n^{T'} (x-t)_+^m$ the approximation obtained by the network of the $n^{th}$ order chebyshev polynomial. Thus

$$S_n^{T'} (x-t)_+^m = \sum_{k=0}^n c_{km}^{'} T_k(x)$$

From equation 17 and the previous equation we get the below

$$|S_n^{T'}(x-t)_+^m - (x-t)_+^m| = |\sum_{k=0}^n c'_{km} T_k(x) - (x-t)_+^m|$$

$$= |\sum_{k=0}^n (c_{km} + \epsilon) T_k(x) - (x-t)_+^m|$$

$$= |\sum_{k=0}^n c_{km} T_k(x) - (x-t)_+^m + \sum_{k=0}^n \epsilon T_k(x)|$$

$$= |\sum_{k=0}^\infty c_{km} T_k(x) - (x-t)_+^m + \sum_{k=0}^n \epsilon T_k(x) - \sum_{k=n+1}^\infty c_{km} T_k(x)|$$

$$= |\sum_{k=0}^n \epsilon T_k(x) - \sum_{k=n+1}^\infty c_{km} T_k(x)|$$

$$\le |\sum_{k=0}^n \epsilon T_k(x)| + |\sum_{k=n+1}^\infty c_{km} T_k(x)|$$

$$= |\sum_{k=0}^n (\mathcal{O}(\frac{C_f^2}{h} + \frac{hn}{N} log(N))) T_k(x)| + |\sum_{k=n+1}^\infty c_{km} T_k(x)|$$

$$= \mathcal{O}(\frac{nC_f^2}{h} + \frac{hn^2}{N} log(N)) + |\sum_{k=n+1}^\infty c_{km} T_k(x)|$$

We now bound the expression $|\sum_{k=n+1}^\infty c_{km} T_k(x)|$. We note that

$$c_{km} = \frac{2}{\pi} \int_t^1 \frac{(x-t)^m T_k(x)}{\sqrt{1-x^2}} dx$$

Using $x = \cos\theta$ and $t = \cos\phi$ we get

$$c_{km} = \frac{2}{\pi} \int_0^\phi (\cos\theta - \cos\phi)^m \cos k\theta d\theta$$

We now have to solve the integral $I = \int_0^\phi (\cos\theta - \cos\phi)^m \cos k\theta d\theta$ We do this by integrating by parts

$$I = [(\cos\theta - \cos\phi)^m \frac{\sin k\theta}{k}]_0^\phi - \int_0^\phi m(\cos\theta - \cos\phi)^{m-1}(-\sin\theta)\frac{\sin k\theta}{k} d\theta$$

$$= -\int_0^\phi m(\cos\theta - \cos\phi)^{m-1}(-\sin\theta)\frac{\sin k\theta}{k} d\theta$$

$$= -\int_0^\phi m(\cos\theta - \cos\phi)^{m-1}\frac{\cos(k-1)\theta + \cos(k+1)\theta}{k} d\theta$$

$$= -(\int_0^\phi m(\cos\theta - \cos\phi)^{m-1}\frac{\cos(k-1)\theta}{k} d\theta + (\int_0^\phi m(\cos\theta - \cos\phi)^{m-1}\frac{\cos(k+1)\theta}{k} d\theta)$$

$$= -(I_{11} + I_{12})$$

Continuing in this manner we get $\mathcal{O}(2^m)$ integrals, one of which is as below

$$I_{m1} = \frac{m(m-1)(m-2)\dots 1}{k(k-1)(k-2)\dots(k-m)}(-\sin\theta)\sin(k-m)\theta$$

$$= \mathcal{O}(m^m k^{-m})$$

$$= \mathcal{O}(k^{-m}) \text{as } k \to \infty$$

Thus $I$ evaluates to $\mathcal{O}(k^{-m})$ and $c_{km} = \mathcal{O}(k^{-m})$.

Thus we have,

$$| \sum_{k=n+1}^{\infty} c_{km} T_k(x)| = \sum_{k=n+1}^{\infty} |c_{km}|$$
$$= \mathcal{O}(n^{-m})$$

Using this result in the expression for $|S_n^{T'}(x-t)_+^m - (x-t)_+^m|$ we get

$$|S_n^{T'}(x-t)_+^m - (x-t)_+^m| = \mathcal{O}(\frac{nC_f^2}{h} + \frac{hn^2}{N}log(N)) + |\sum_{k=n+1}^{\infty} c_{km}T_k(x)|$$
$$= \mathcal{O}(\frac{nC_f^2}{h} + \frac{hn^2}{N}log(N)) + \mathcal{O}(n^{-m})$$

Finally using equations 16 and 15 for $S_n^{T'}G$ we get

$$|S_n^{T'}G - G| = \mathcal{O}(\frac{nC_f^2}{h} + \frac{hn^2}{N}log(N) + n^{-m}) \tag{18}$$

which concludes the proof.

$\square$

The bound states that as the filter order $n$ and the hidden dimension of the network $h$ are increased (subject to $n^3C_f^2 \leq hn^2 \leq \frac{N}{\log N}$) the approximation will converge to the desired filter response. The condition can be thought to satisfy the statistical rule that the model parameters must be less than the sample size. In the limit of $N \rightarrow \infty$, $h$ and $n$ could be increased to as large values as desired subject to $h \geq \mathcal{O}(n)$, which is the order of parameters to be approximated. This is equivalent to say that if we do not consider the generalization error, we can theoretically take a large $h$. Then, we are left with only the term containing the filter order i.e. the approximation error comes down to $\mathcal{O}(n^{-m})$ as expected. One point to note is that we assume suitable coefficients $c_{km}$ can be learned from the input graph. This assumption requires that the graph has the necessary information regarding spatial connectivity and signals. In the current implementation, we only use the information from the signals residing on the graph nodes in the attention heat map and discard the spatial connectivity. It is a straightforward exercise to extend this idea by using multi-relational graphs to include the graph signals and the spatial connectivity information which we leave for future works. Nevertheless, the current implementation does well empirically, as is evident from the results on the real world and synthetic (A.4) datasets.

### A.1.1 TRANSFORMERS AND WL TEST

WL-test has been used as a standard measure to study the expressivity of GNNs. The k-WL test is the variant of the WL test that works on k-tuples instead of one-hop node neighbors compared to the standard 1-WL test. Recently, with the rapid adoption of transformers on graph tasks, the equivalence of transformer and WL test naturally arises. In the following section, we try to argue how a transformer can approximate the WL test.

Given a sequence, recent works by (Yun et al., 2019; 2020) have theoretically illustrated that Transformers are universal sequence-to-sequence approximators. The core building block of the transformer is a self-attention layer; the self-attention layers compute dynamic attention values between the query and key vectors by attending to all the sequences. This can be viewed as passing messages between all nodes, regardless of the input graph connectivity.

For position encoding, recently, many works have tried using eigenvectors and eigenvalues as PEs for GNNs (Dwivedi & Bresson, 2020; Kreuzer et al., 2021). The recent work by DGN (Beaini et al., 2021) shows how using eigenvalues can distinguish non-isomorphic graphs which WL test cannot.

Transformers are universal approximators coupled with eigenvalues and eigenvectors as position encodings are powerful than the WL test given enough model parameters. However, they can only

| DATASET | GPU | Memory | TIME (sec PER EPOCH) |
|---------|-----|--------|----------------------|
| MUTAG | Geforce P8 | 8 | 1.5 |
| NCI1 | Geforce P8 | 8 | 37.8 |
| Molhiv | Tesla V100 | 16 | 79.0 |
| PATTERN | Tesla V100 | 16 | 104.2 |
| CLUSTER | Tesla V100 | 16 | 115.4 |
| ZINC | Tesla V100 | 16 | 529.9 |

Table 4: Computational details used for the datasets on the SpecTRA-Base setting

| Model | Dataset | PE | PE layers | PE dim | no. layers | hidden dim | Model Params | Heads | Filter Order |
|-------|---------|----|-----------|--------|-----------|-----------|-------------|-------|-------------|
| SpecTRA-Base | MUTAG | - | - | - | 3 | 64 | 118074 | 4 | 8 |
| | NCI1 | - | - | - | 3 | 64 | 122194 | 4 | 8 |
| | Molhiv | - | - | - | 3 | 64 | 129465 | 4 | 8 |
| | PATTERN | - | - | - | 3 | 64 | 5465930 | 4 | 8 |
| | CLUSTER | - | - | - | 3 | 64 | 5467726 | 4 | 8 |
| | ZINC | - | - | - | 3 | 64 | 351537 | 4 | 8 |
| Vanilla-Transformer | MUTAG | - | - | - | 3 | 64 | 119080 | 4 | 8 |
| | NCI1 | - | - | - | 3 | 64 | 107074 | 4 | 8 |
| | Molhiv | - | - | - | 3 | 64 | 127356 | 4 | 8 |
| | PATTERN | - | - | - | 3 | 64 | 5445389 | 4 | 8 |
| | CLUSTER | - | - | - | 3 | 64 | 5447657 | 4 | 8 |
| | ZINC | - | - | - | 3 | 64 | 340737 | 4 | 8 |

Table 5: Model architecture parameters of SpecTRA-Base and Vanilla-Transformer

approximate the solution to the graph isomorphism problem with a specific error and not solve them fully which is also proven by Kreuzer et al. (2021). The same holds for us as we inherit the identical characteristics from SAN.

## A.2 DATASET DETAILS

We benchmark the widely used datasets for graph classification, node classification, and graph regression. Namely for graph classification we use MUTAG (Morris et al., 2020), NCI1 (Morris et al., 2020) and, the ogbg-MolHIV (Hu et al., 2020) dataset, for node classification we use the PATTERN and CLUSTER datasets (Dwivedi et al., 2020) and for graph regression we run our method on the ZINC (Dwivedi et al., 2020) dataset.

MUTAG is a collection of nitroaromatic compounds. Here, the goal is to predict the mutagenicity of these compounds. Input graphs represent the compounds with atoms as the vertices and bonds as the edges. Similarly, in NCI1, the graphs represent chemical compounds with nodes representing the atoms and the edges indicating their bonds. The atoms are labeled as one-hot vectors as node features. Ogbg-MolHIV is a molecular dataset in which each graph consists of a molecule, and the nodes have their features encoded as the atomic number, chirality, and other additional features. PATTERN and CLUSTER are node classification datasets constructed using stochastic block models. In PATTERN, the task is to identify subgraphs and CLUSTER aims at identifying clusters in a semi-supervised setting. ZINC is a database of commercially available compounds. The task is to predict the solubility of the compound formulated as a graph regression problem. Each molecule has the type of heavy atom as node features and the type of bond as edge features.

## A.3 EXPERIMENT SETTINGS

Table 4 lists the hardware and the run time of the experiments on each dataset. Table **??** lists out the model architecture parameters for each configurations. For the configurations *SpecTRA-Base*, *SpecTRA+LapE*, *SpecTRA+3RW* and *SpecTRA+GCKN+3RW* we used the default hyper-parameters provided by GraphiT (Mialon et al., 2021) as we inherited position encodings from GraphiT. For instance, in ZINC, we don't use edge features. In the configurations *SpecTRA+LPE+Sparse* and *SpecTRA+LPE+Full* we used the configurations from SAN (Kreuzer et al., 2021). This is to ensure same experiment settings which these encoding schemes have used while inducing position encodings in the vanilla transformer models. Means and uncertainties are derived from four runs

| Model | Dataset | PE | PE layers | PE dim | no. layers | hidden dim | Model Params | Heads | Filter Order |
|---|---|---|---|---|---|---|---|---|---|
| SpecTRA + LapE | MUTAG | LapE | 1 | 64 | 3 | 64 | 2216402 | 4 | 8 |
| | NCI1 | LapE | 1 | 64 | 3 | 64 | 2218322 | 4 | 8 |
| | Molhiv | LapE | 1 | 64 | 3 | 64 | 129657 | 4 | 8 |
| | PATTERN | LapE | 1 | 64 | 3 | 64 | 4414026 | 4 | 8 |
| | CLUSTER | LapE | 1 | 64 | 3 | 64 | 5468494 | 4 | 8 |
| | ZINC | LapE | 1 | 64 | 3 | 64 | 483401 | 4 | 8 |
| GraphiT + LapE | MUTAG | LapE | 1 | 64 | 3 | 64 | 2195467 | 4 | 8 |
| | NCI1 | LapE | 1 | 64 | 3 | 64 | 2207266 | 4 | 8 |
| | Molhiv | LapE | 1 | 64 | 3 | 64 | 127489 | 4 | 8 |
| | PATTERN | LapE | 1 | 64 | 3 | 64 | 4395278 | 4 | 8 |
| | CLUSTER | LapE | 1 | 64 | 3 | 64 | 5447384 | 4 | 8 |
| | ZINC | LapE | 1 | 64 | 3 | 64 | 474456 | 4 | 8 |
| SpecTRA + 3RW | MUTAG | RW | 1 | - | 3 | 64 | 106694 | 4 | 8 |
| | NCI1 | RW | 1 | - | 3 | 64 | 110698 | 4 | 8 |
| | Molhiv | RW | 1 | - | 3 | 64 | 129465 | 4 | 8 |
| | PATTERN | RW | 1 | - | 3 | 64 | 4413258 | 4 | 8 |
| | CLUSTER | RW | 1 | - | 3 | 64 | 5467726 | 4 | 8 |
| | ZINC | RW | 1 | - | 3 | 64 | 482825 | 4 | 8 |
| GraphiT + 3RW | MUTAG | RW | 1 | - | 3 | 64 | 104578 | 4 | 8 |
| | NCI1 | RW | 1 | - | 3 | 64 | 106498 | 4 | 8 |
| | Molhiv | RW | 1 | - | 3 | 64 | 125745 | 4 | 8 |
| | PATTERN | RW | 1 | - | 3 | 64 | 4394786 | 4 | 8 |
| | CLUSTER | RW | 1 | - | 3 | 64 | 5445478 | 4 | 8 |
| | ZINC | RW | 1 | - | 3 | 64 | 338817 | 4 | 8 |
| SpecTRA + GCKN + 3RW | MUTAG | GCKN+RW | 1 | 32 | 3 | 64 | 116783 | 4 | 8 |
| | NCI1 | GCKN+RW | 1 | 32 | 3 | 64 | 2228434 | 4 | 8 |
| | Molhiv | GCKN+RW | 1 | 32 | 3 | 64 | 5619529 | 4 | 8 |
| | PATTERN | GCKN+RW | 1 | 32 | 3 | 64 | 37988386 | 4 | 8 |
| | CLUSTER | GCKN+RW | 1 | 32 | 3 | 64 | 37990182 | 4 | 8 |
| | ZINC | GCKN+RW | 1 | 32 | 3 | 64 | 499273 | 4 | 8 |
| GraphiT + GCKN + 3RW | MUTAG | GCKN+RW | 1 | 32 | 3 | 64 | 114882 | 4 | 8 |
| | NCI1 | GCKN+RW | 1 | 32 | 3 | 64 | 2205672 | 4 | 8 |
| | Molhiv | GCKN+RW | 1 | 32 | 3 | 64 | 5598726 | 4 | 8 |
| | PATTERN | GCKN+RW | 1 | 32 | 3 | 64 | 37889986 | 4 | 8 |
| | CLUSTER | GCKN+RW | 1 | 32 | 3 | 64 | 37895163 | 4 | 8 |
| | ZINC | GCKN+RW | 1 | 32 | 3 | 64 | 478645 | 4 | 8 |

Table 6: Model architecture parameters of SpecTRA with position embedding from GraphiT and original GraphiT model

| Model | Dataset | PE | PE layers | PE dim | no. layers | hidden dim | Model Params | Heads | Filter Order |
|---|---|---|---|---|---|---|---|---|---|
| SpecTRA + LPE + Sparse | MUTAG | LPE | 1 | 16 | 6 | 64 | 558322 | 4 | 8 |
| | NCI1 | LPE | 1 | 16 | 6 | 64 | 559762 | 8 | 8 |
| | Molhiv | LPE | 2 | 16 | 6 | 96 | 608129 | 4 | 8 |
| | PATTERN | LPE | 3 | 16 | 6 | 96 | 579013 | 10 | 8 |
| | CLUSTER | LPE | 1 | 16 | 16 | 56 | 412978 | 8 | 8 |
| | ZINC | LPE | 3 | 16 | 6 | 96 | 364167 | 8 | 8 |
| SAN + LPE + Sparse | MUTAG | LPE | 1 | 16 | 6 | 64 | 542082 | 4 | 8 |
| | NCI1 | LPE | 1 | 16 | 6 | 64 | 543522 | 8 | 8 |
| | Molhiv | LPE | 2 | 16 | 6 | 96 | 602672 | 4 | 8 |
| | PATTERN | LPE | 3 | 16 | 6 | 96 | 570736 | 10 | 8 |
| | CLUSTER | LPE | 1 | 16 | 16 | 56 | 403783 | 8 | 8 |
| | ZINC | LPE | 3 | 16 | 6 | 96 | 360617 | 8 | 8 |
| SpecTRA + LPE + Full | MUTAG | LPE | 1 | 16 | 6 | 64 | 640242 | 4 | 8 |
| | NCI1 | LPE | 1 | 16 | 6 | 64 | 641682 | 8 | 8 |
| | Molhiv | LPE | 2 | 16 | 6 | 96 | 732984 | 4 | 8 |
| | PATTERN | LPE | 3 | 16 | 6 | 96 | 697003 | 10 | 8 |
| | CLUSTER | LPE | 1 | 16 | 16 | 56 | 867046 | 8 | 8 |
| | ZINC | LPE | 3 | 16 | 6 | 96 | 458303 | 8 | 8 |
| SAN + LPE + Full | MUTAG | LPE | 1 | 16 | 6 | 64 | 624002 | 4 | 8 |
| | NCI1 | LPE | 1 | 16 | 6 | 64 | 625442 | 8 | 8 |
| | Molhiv | LPE | 2 | 16 | 6 | 96 | 714769 | 4 | 8 |
| | PATTERN | LPE | 3 | 16 | 6 | 96 | 688534 | 10 | 8 |
| | CLUSTER | LPE | 1 | 16 | 16 | 96 | 858538 | 8 | 8 |
| | ZINC | LPE | 3 | 16 | 6 | 96 | 454753 | 8 | 8 |

Table 7: Parameters of SpecTRA with position embedding from SAN compared with original SAN

| Method | MUTAG | NCI1 | ZINC | PATTERN | CLUSTER | ogbg-molhiv |
|---|---|---|---|---|---|---|
| Avg $|\mathcal{V}|$ | 30.32 | 29.87 | 23.15 | 118.89 | 117.20 | 25.51 |
| Avg $|\mathcal{E}|$ | 32.13 | 32.30 | 24.90 | 3039.28 | 2,150.86 | 27.47 |
| Node feature | L | L | A | L | L | A |
| Dim(feat) | 38 | 37 | 28 | 3 | 6 | 9 |
| #Classes | 2 | 2 | NA | 2 | 6 | 2 |
| #Graphs | 4,127 | 4,110 | 250,000 | 14,000 | 12,000 | 41,127 |

Table 8: Dataset statistics (L indicates node categorical features and A denotes node attributes).

with different seeds, same as SAN. Additionally, with the final optimized parameters, we reran 10 experiments with identical seeds, which is same as SAN's/Mialon et al/Dwivedi et al's experiment settings. Vanilla transformer implementation and its values in main paper is taken from Mialon et al. (2021).

### A.4 SYNTHETIC DATASET

To show the benefit of graph-specific filtering compared to static filtering, we run experiments on synthetic datasets that have different spectral components for different graphs. We begin by noting a few basic properties of the Laplacian and its spectral decomposition before detailing the data generation process. The below equation gives the unnormalized Laplacian ($L = D - A$) of a graph:

$$L(i, j) = \begin{cases} deg(i) & \text{if } i = j \\ -1 & \text{if } (i, j) \in E \\ 0 & \text{otherwise} \end{cases}$$

where $deg(i)$ is the degree of the node $i$ and $E$ is the set of edges in the graph. Multiplying $L$ by a vector $v$ gives the below expression

$$w = Lv$$
$$w(i) = \sum_{i,j \in E} (v(i) - v(j))$$

The expression $v^T L v$ gives

$$v^T L v = \sum_i v(i) \sum_{i,j \in E} (v(i) - v(j))$$
$$= \sum_i \sum_{i,j \in E} v(i)(v(i) - v(j))$$
$$= \sum_{j>i,(i,j) \in E} v(i)(v(i) - v(j)) + v(j)(v(j) - v(i))$$
$$= \sum_{j>i,(i,j) \in E} (v(i) - v(j))^2$$

Thus we can see that the expression $v^T L v$ evaluates to the sum of squared distances between neighboring nodes in the graph. We also note that due to this property, the laplacian is a positive semi-definite matrix. If $v$ were the eigenvector of the graph, we know that $v^T L v$ would be the eigenvalue corresponding to that vector by the spectral decomposition theory. Thus all the eigenvalues of the laplacian are non-negative.

We now try to develop an intuition of the laplacian's lowest and largest eigenvalues/vectors. We see from the above equations that $v^T L v$ is the sum of squared differences between values on the nodes. Thus the smallest eigenvalue would correspond to the eigenvector that assigns the same value to all the neighboring nodes, subject to $\|v\| = 1$. The second eigenvalue would correspond to the orthonormal vector to the first vector and minimizes the sum of squared differences of nodes within a cluster. Thus the second eigenvector would try to keep values of its components similar/closer for

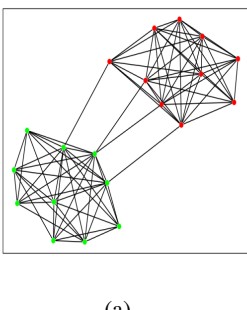 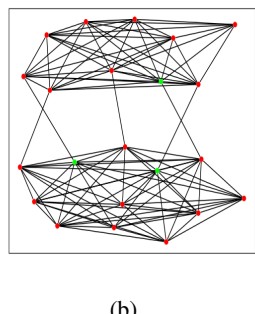 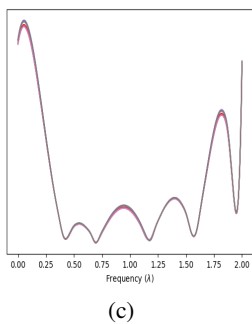

(a) (b) (c)

Figure 4: Graphs and Filter Frequency response for the graphs on the $Synthetic_1$ dataset for the case with $K = 4$ and $h = 1$. Figure (a) is a graph that illustrates low frequency signals (here, we observe the neat clusters as expected for low frequency information). Figure (b) represents a graph with a high frequency component (we observe the mixing of signals in nodes of the same cluster, i.e., different intra-cluster signal values). Finally, figure (c) is the aggregated frequency response across the dataset with the normalized frequency along the X-axis and associated magnitudes on the Y-axis. Here we note the filter has learned the two components of the graph spectrum: the low frequency and the high frequency components.

nodes that belong to the same cluster. Similarly, the vector corresponding to the largest eigenvalue would try to maximize the sum of squared differences between neighboring nodes and have its nodes(components) belonging to the same take different values subject to $\|v\| = 1$ while also being orthonormal to the other vectors.

With the above background, we generate synthetic datasets with node signals exhibiting different spectral components for different graphs. Consider $L$ to be the graph laplacian with eigenvalues $\lambda$ and eigenvectors $V$. We generate stochastic block matrices (SBMs) with $B$ number of blocks and $N$ number of nodes per block having an intracluster density of edges as $p_i$ and the inter-cluster edge density as $p_o$. The signals are assigned to the graphs according to the eigenvectors of the selected components of the spectrum. To keep it simple, we use only a single eigenvector ($V_i$) to generate signals per graph. Specifically, out of the $N_G = \mathcal{E}(NB)$ eigenvalues, we select one and take the components of the corresponding eigenvectors. The nodes are then clustered using standard clustering algorithms, such as K-means, into $C$ classes that we keep equal to the number of blocks. Each node is assigned a one-hot vector at the position of the class it belongs to. We then drop this attribute of $50\%$ of the nodes i.e., these nodes are assigned a $0$ vector, and the task is to find the correct assignment for the unknown class based on the connectivity pattern and the spectrum of the known signals on the graph. Thus this task boils down to finding the suitable spectral component in the graph signal and using this information for classification. This should benefit from a graph-specific decomposition and filtering approach, which we confirm from the empirical results. We generate 3 datasets namely $Synthetic_1$, $Synthetic_2$, and $Synthetic_3$ using different values of $N$, $B$, $p_i$, $p_o$, $V_i$. $Synthetic_1$ has $N = 10$, $B = 2$, $p_i = 0.9$, $p_o = 0.05$, $\{V_i \mid i \in \{1, N_G\}\}$, with 1000, 100, 100 graphs in the train, test and valid graphs respectively. $Synthetic_2$ has $N = 10$, $B = 6$, $p_i = 0.9$, $p_o = 0.05$, $\{V_i \mid i \in \{1, N_G\}\}$, with 1000, 100, 100 graphs in the train, test and valid graphs respectively. $Synthetic_1$ has $N = 10$, $B = 6$, $p_i = 0.9$, $p_o = 0.05$, $\{V_i \mid i \in \{1, \lceil \frac{N_G}{2} \rceil, N_G\}\}$, with 1000, 100, 100 graphs in the train, test and valid graphs respectively.

We train the synthetic datasets on the SpecTRA-Static and SpecTRA-Base models to study the effect of graph-specific dynamic filters on the graph. For the $Synthetic_1$ dataset, we use the hidden dimension as 16, and for $Synthetic_2$ and $Synthetic_3$ we keep it to 64. The number of layers is fixed to 1. The number of heads $h$ and filter order $K$ for SpecTRA-Static are kept at 1 and 4 respectively and for varied for other settings as can be seen in table A.4 . We can see from table A.4 that the performance on the synthetic datasets, using dynamic filters of SpecTRA-Base, has increased by a large margin as compared to the case of static dataset-specific filters in SpecTRA-Static using the same model parameters. This justifies the benefit and necessity of graph-specific filter design in

| Model | | | $Synthetic_1$ | $Synthetic_2$ | $Synthetic_3$ | #Param(dim=16) | #Param(dim=64) |
|---|---|---|---|---|---|---|---|
| SpecTRA-Static | $K=4$ | $h=1$ | $70.04 \pm 4.41$ | $35.34 \pm 0.25$ | $33.05 \pm 0.48$ | 4122 | 62958 |
| SpecTRA-Base | $K=2$ | $h=1$ | $89.67 \pm 0.50$ | $45.63 \pm 0.30$ | $39.67 \pm 0.55$ | 3582 | 54738 |
| | $K=4$ | | $92.15 \pm 0.20$ | $46.32 \pm 0.42$ | $40.22 \pm 0.61$ | 4122 | 62958 |
| | $K=8$ | | $92.26 \pm 0.40$ | $47.14 \pm 0.63$ | $40.27 \pm 0.30$ | 5250 | 79446 |
| SpecTRA-Base | $K=2$ | $h=4$ | $79.26 \pm 7.47$ | $46.26 \pm 0.20$ | $39.83 \pm 0.33$ | 3090 | 47010 |
| | $K=4$ | | $91.54 \pm 0.56$ | $46.59 \pm 0.34$ | $39.42 \pm 0.30$ | 3150 | 47550 |
| | $K=8$ | | $92.35 \pm 0.37$ | $46.24 \pm 0.56$ | $40.74 \pm 0.35$ | 3318 | 48678 |
| SpecTRA-Base | $K=2$ | $h=8$ | $74.99 \pm 0.43$ | $45.88 \pm 0.50$ | $38.60 \pm 1.05$ | 3064 | 46618 |
| | $K=4$ | | $91.32 \pm 0.43$ | $45.77 \pm 0.73$ | $38.86 \pm 0.17$ | 3100 | 46774 |
| | $K=8$ | | $91.60 \pm 0.30$ | $45.75 \pm 0.73$ | $39.71 \pm 0.50$ | 3220 | 47134 |

Table 9: Study on the performance of SpecTRA-Base vs SpecTRA-Static on the synthetic datasets. We study the effect of the order $K$ of filters and the number of heads $h$ for SpecTRA-Base.

cases where the spectral information differs from graph to graph. We also observe that as the filter order is increased for a given number of heads, the performance improves.

On the other hand, lower filter order is detrimental to the task. The performance saturates if the filter order is increased beyond a specific limit, as is evident from the $Synthetic_1$ dataset. Also, we do not observe any improvement by increasing the number of heads, keeping the filter order fixed, in this case. This may be due to the nature of the dataset, where we have restricted each graph to contain a single spectral component. We leave it to future studies to determine the effect of number of heads on multiple spectral components in the signal.

### A.5 FILTER FREQUENCY RESPONSE ON OTHER DATASETS

Here we provide the plots of the frequency response of the filters learned on the other datasets which has not been included in the main paper due to page limit.

**Filter Frequency response on ZINC**: The frequency response of the filters learned on ZINC with filter order four is given in figure 5. Each curve is the frequency response of a filter learned for a single head. We see various filters being learned, such as few low pass, high pass, and band stop filters. Figure 6 shows the frequency response for filters of order 8. Here we observe the high pass and multi-modal band pass responses being learned.

**Filter Frequency response on MolHIV**: Figure 7 illustrates the frequency response for filters learned on the MolHIV. Here, we observe the low pass and multi-modal band pass filter responses.

**Filter Frequency response on PATTERN**: Figure 8 shows the frequency response for filters learned on the PATTERN dataset. Here we observe the filters allowing signals in the low and high frequency regime along with some magnitude assigned to the mid-frequency components. The filter order eight (the two right-most plots) shows a surprising result of visually indistinct filters being learned despite the regularization. This indicates the task has a high bias towards signals in the low-frequency region and some components in the middle and high frequency regime to an extent. This could be interpreted as the model trying to learn the two components: the SBM, which corresponds to the low-frequency signal, and the underlying pattern itself, which may benefit from the middle and high frequency components in the spectrum.

**Filter Frequency response on CLUSTER**: Similar to PATTERN, the filter response plots for CLUSTER as in figure 9 show low-frequency filters being learned along with bandpass filters. This is intuitive as the task of CLUSTER would indeed benefit by learning low-frequency signals for grouping nodes belonging to the same cluster. All these observations across all datasets validates our hypothesis studied in the scope of this paper.

### A.6 INTERPRETABILITY IN SPECTRAL SPACE

In this section, we look at the interpretability aspect induced by SpecTRA in the spectral space. The figures 10 and 11 show the graphs with the eigenvectors on the nodes corresponding to the top eigenvalues selected by the filter. From both figures, we see that in the case of the low pass filter (leftmost graph in sub-figure (a), (d) of figure 10 and subfigure (a)-(d) of figure 11), the nodes

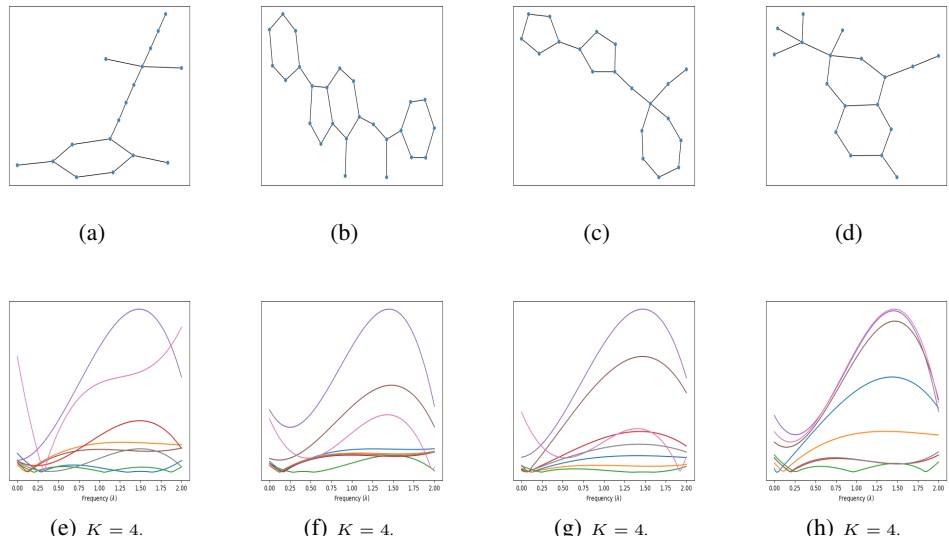

Figure 5: Filter Frequency response on individual graphs on the ZINC dataset for a filter order of 4. Figures (a) ∼ (d) are the graphs from the dataset and Figures (e) ∼ (g) are the corresponding frequency responses. X axis shows the normalized frequency with magnitudes on the Y axis.

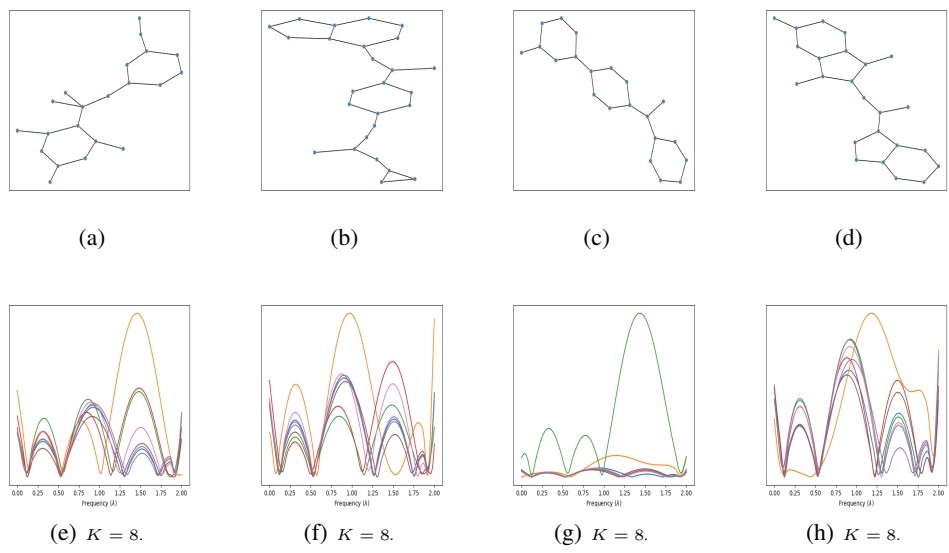

Figure 6: Filter Frequency response on individual graphs on the ZINC dataset for a filter order of 8. Figures (a) ∼ (d) are the graphs from the dataset and Figures (e) ∼ (g) are the corresponding frequency responses. X axis shows the normalized frequency with magnitudes on the Y axis.

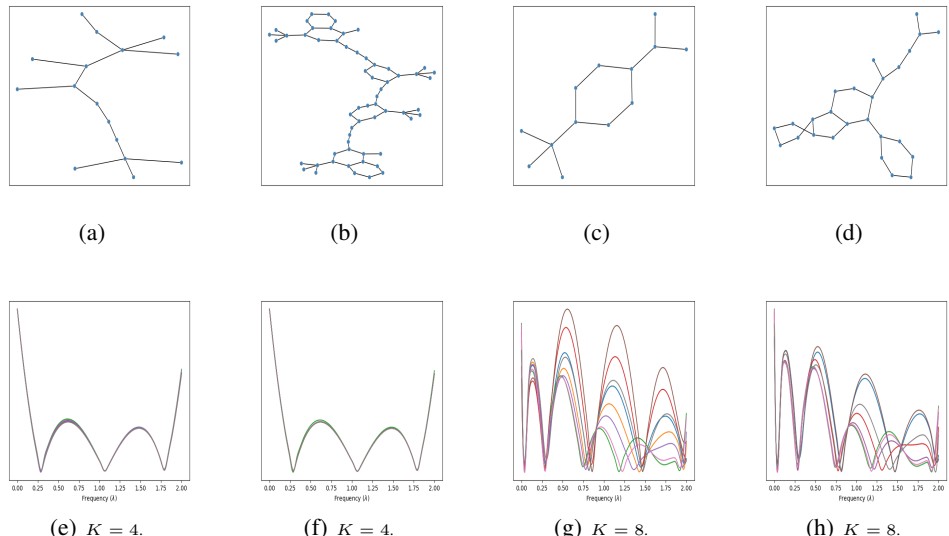

Figure 7: Filter Frequency response on individual graphs on the MolHIV dataset. Figures (a) ∼ (d) are the graphs from the dataset and Figures (e) ∼ (h) are the corresponding frequency responses. X axis shows the normalized frequency with magnitudes on the Y axis.

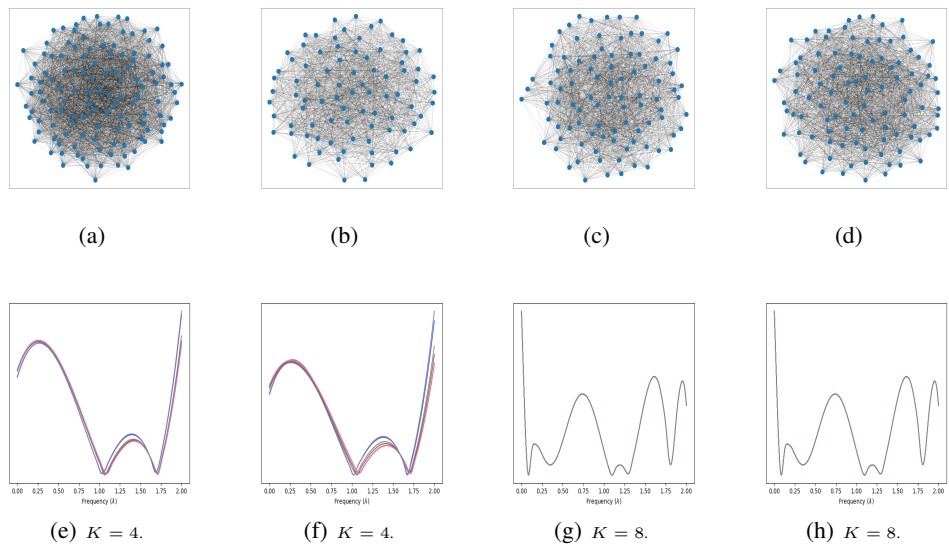

Figure 8: Filter Frequency response on individual graphs on the PATTERN dataset. Figures (a) ∼ (d) are the graphs from the dataset and Figures (e) ∼ (h) are the corresponding frequency responses. X axis shows the normalized frequency with magnitudes on the Y axis.

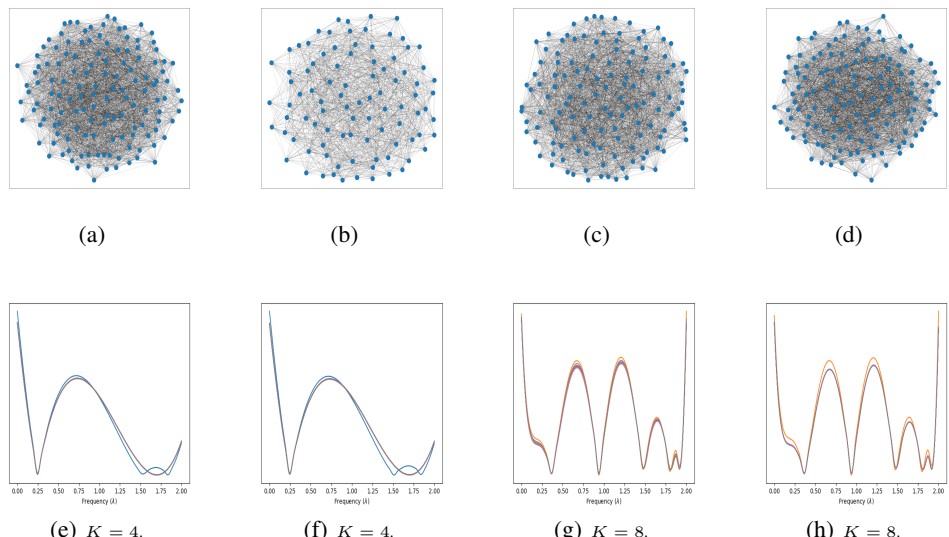

Figure 9: Filter Frequency response on individual graphs on the CLUSTER dataset. Figures (a) ~ (d) are the graphs from the dataset and Figures (e) ~ (g) are the corresponding frequency responses. X axis shows the normalized frequency with magnitudes on the Y axis.

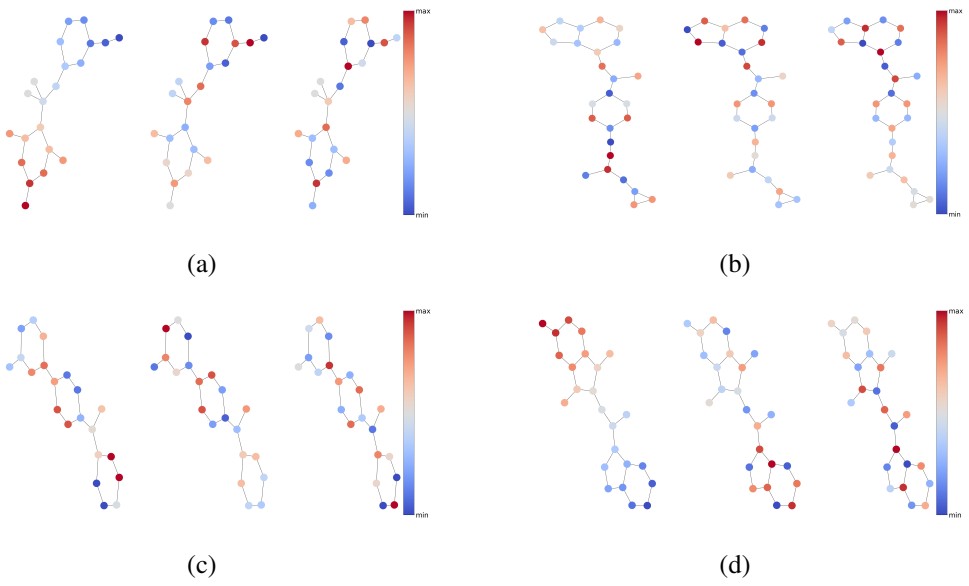

Figure 10: Attention heat map in spectral space of the sample graphs in ZINC dataset determined from the frequency response in Figure 6 for its each sub-graph (a)-(d). Blue illustrates the lower end of the spectrum and red color shows the higher end of the spectrum.

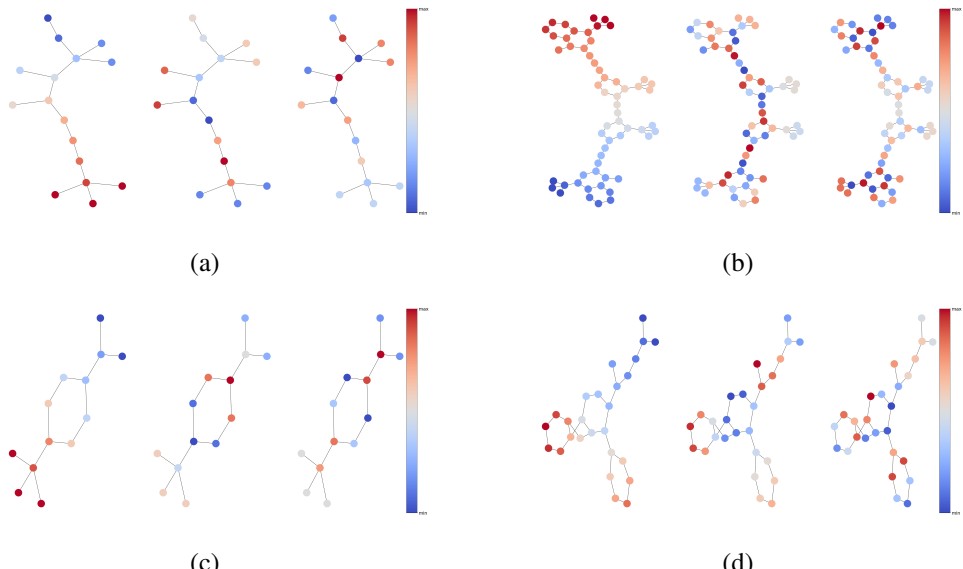

(a)                      (b)

(c)                      (d)

Figure 11: Attention heat map in spectral space of the sample graphs in MolHIV dataset determined from the frequency response in Figure 7 for its each sub-graph (a)-(d). Blue illustrates the lower end of the spectrum and red color shows the higher end of the spectrum.

in the neighborhood forming a cluster have similar eigenvalues. Whereas in the highpass filter ( (rightmost graph in each subfigure (a)-(d))) the eigenvalues of the nodes alternate, with neighboring nodes taking distinct values and far off nodes having similar values.

We could make two interpretations of this phenomenon. The first one is closely related to *attention* in the spatial space where SpecTRA attends to select input features. In this case, we can think of the model learning to pay more attention to the nodes with higher eigenvalues and lesser attention to the nodes with smaller eigenvalues in a graph and task-specific manner. The second interpretation is related to the *interaction* between the nodes, i.e., for a given node which other nodes are considered for aggregation. For example, in graph attention networks, the neighboring nodes are aggregated, and the values of nodes in the same cluster tend to be closer to each other (homophily). This is a particular case of the low pass filter in which we can see from the figures 10 and 11 that the nodes belonging to the same cluster take on similar values. For example, consider the Figure 10 (d). The leftmost graph has nodes in a particular cluster taking similar eigenvalues showing short-range dependencies (interactions). In the rightmost graph, nodes in the same clusters take on different eigenvalues, illustrating the need for long-range interactions. However, depending on the task and graph, the model also learns to aggregate (interact with) distant nodes, as seen in the graph corresponding to the high pass filter (the rightmost graphs in each sub-figure of 10 and 11). Thus SpecTRA can be interpreted as a generic(covering the entire spectrum of frequencies) *attention* network in the *spectral space*.

