# OpenReview forum: "SpecTRA: Spectral Transformer for Graph Representation Learning"
_ICLR.cc/2022/Conference — ICLR 2022 Submitted_

### Official Review · Reviewer_VYTd · 2021-10-31

**Correctness:** 4
**Technical Novelty And Significance:** 3
**Empirical Novelty And Significance:** 2
**Recommendation:** 5
**Confidence:** 4

**Main Review:**

Correctness: No correcteness concerns.

Technical novelty and significance: The idea that different filter shapes are necessary to discriminate the spectral components associated with different graphs makes sense, therefore, learning the filter coefficients of the GNN used to distinguish between these graphs is a compelling solution. However, other architectural choices are not as well motivated. For example, why do the authors learn the spectral GCN filter weights from the attention matrix of the transformer, which can have a completely different sparsity pattern than the input graph? Why not learn the filter weights from the graph itself, e.g., by using a GNN? What is the rationale behind this choice of architecture?
Another technical concern relates to Theorem 4.1. While the authors claim that this theorem justifies the proposed architecture, it is not clear how. That smooth functions can be approximated by Chebyshev polynomials is a well-known result. Can the authors clarifiy which aspects or parts of their method are justified by Theorem 4.1? I recommend adding a more detailed discussion of the theorem's implications to Section 4.1.

Empirical novelty and significance: One of the goals of the numerical results is to assess "the impact of recently proposed position encoding schemes on SpecTra". Yet, none of these position encoding schemes were proposed by the authors. Moreover, the numerical results are unconvincing. The proposed method achieves similar performance to the compared methods except for ZINC. Additionally, errors with three decimal digits are not statistically significant for averages over 4 runs.

Other comments:
- The abstract and the introduction are difficult to understand and do not give a clear picture of the paper's contributions.
- In the beginning of Section 4.1, the authors should specify how the node feature embeddings fed to transformer are obtained from the graph.

**Summary Of The Paper:**

This paper proposes a model for graph representation learning based on a transformer and on two GNNs: a message-passing neural network that learns graph filter coefficients from the transformer's attention weights, and a spectral GCN with parameters given by these filters coefficients. This choice of architecture is motivated by the assumption that different filter shapes are necessary to discriminate the specific spectral components associated with different graphs.

**Summary Of The Review:**

The idea that different filter shapes are necessary to discriminate the spectral components associated with different graphs makes sense, therefore, learning the filter coefficients of the GNN used to distinguish between these graphs is a compelling solution. However, other architectural choices---such as learning the spectral GCN filter weights from the transformer's attention matrix---are not well motivated. Moreover, the numerical results are unconvincing.

---

> ### Author Response · Authors · 2021-11-14
> **Thank you and author's response to reviewer's comments**
>
> We sincerely thank the reviewer for taking time out of busy schedule and voluntarily spending time reviewing our work.
>   - Question: Introduction and Abstract:
>     - Answer: Many thanks. We have re-written the abstract and modified the introduction in rebuttal. Some experiment settings have been moved to the appendix plus  few equations were combined to fix space issues.
>     - As advised, we added discussion in section 4.1 regarding theorem and state the motivation of our choices in architecture in rebuttal-version section 4.1.
>
> - Question: Reg. Theorem 4.1
>     - Answer: This theorem argues that the chebyshev polynomials are able to learn any smooth function in the domain [-1,1]. Using injective aggregators in GNNs and universal approximators in MLP we could approximate these coefficients to the desired precision. This motivates the use of the GNNs followed by MLPs to learn the filter coefficients. The input to the GNNs could be divided into two parts: the input graph indicating the connectivity of nodes and the signals on the graph. The signals residing on the nodes are important in determining the frequency and this is the aspect we wanted to study in this work. For example, considering a one dimensional signal we will not be able to determine the frequency components by ignoring the signals. We need to look at the recurrence of the signals, loosely speaking at what intervals it changes sign etc. This, we argue in some senses, can be characterised by the “spatial closeness” given by the attention maps of the transformer module in general graphs. This is the rationale behind using the transformer and its attention weights. Since our aim was to study if the spectral components could be identified by looking at the signals and using the “spatial connectivity pattern” we ignore the connectivity structure imposed by the original graph, which may also be important. To handle this case (for the second type of input to GNN), we do briefly mention in section appendix A.1 below the proof for theorem 4.1 that we could use a heterogenous graph containing the original graph and the “spatial connectivity” obtained from the transformer to learn the filter coefficients. However, as that was not the main aim of the work we left it as improvements for future works.
> - Question: Regarding four runs,
>     - Answer: we do exactly the same as SAN as mean value and uncertainties are derived from four runs with different seeds. Additionally, with the final optimized parameters, we reran 10 experiments with identical seeds, which is same as SAN’s/Mialon et al/Dwivedi et al s experiment settings. We added this comment in the appendix in the rebuttal version while describing parameters.
>
> - Question: Empirical novelty:
>     - Answer: We agree with reviewers' observation that the best values are achieved only after inducing position encoding schemes. Our rationale here is: Is the performance jump of SpecTRA-base observed over vanilla transformers also maintained when inducing position encodings? Position encodings were not our main focus in this work hence we did not propose a new one. We wanted to explore another school of thought that aims to improve transformers to filter out the noise and capture the entire spectrum of graphs. However, when we induce position encoding, behavior is exactly what is expected as a vanilla transformer:
>       - with position encoding schemes of Mialon et al (basically replacing vanilla transformer in Mialon et al implementation), our model improves the SotA values of Mialon et al for 3/6 datasets (NC1, MUTAG, ZINC) where the jump is coming from SpecTRA ability considering everything else is same.   Position encodings of Mialon et al show limited performance on Pattern/Cluster; using their position encoding in SpecTRA has similar behavior on these datasets. With SAN position encodings, achieving comparable (Pattern) or better (on Cluster and MOLHIV) where original SAN is a SotA. It signifies two things:
>         -  Proposed position encoding schemes dont work agnostic of tasks neither for a vanilla transformer nor for its variant such as SpecTRA-Base. Our performance jumps are in the range of the previous baseline's jump on its predecessor.
>         -  SpecTRA, when improving capabilities of the vanilla transformer model, compliments performance jump achieved from position encoding schemes, depending on their original SotA datasets where the vanilla transformer was used as a base. Hence, our results established an important aspect: it is not just the position encoding that helps but also filtering out the noise in a transformer that positively impacts encoding schemes. An ideal scenario for future work is to propose a model that filters out noise plus inherently does position encoding- also pointed out by us in the limitation section.
>
>
> We sincerely hope our data points clarify the reviewer's concern. If yes, would you be kind to us to increase our score?

---

> > ### Author Response · Authors · 2021-11-22
> > **Extended Evaluation using newly proposed position encoding-Authors response part2**
> >
> > Dear Reviewer,
> >
> > We executed new experiments using SpecTRA base plus a newly proposed position encoding (https://openreview.net/pdf?id=wTTjnvGphYj- ICLR2022 submission) to further strengthen our claim that SpecTRA's ability complements the position encoding behavior. Our new experiments (please see common comment to all reviewers) also justify why we did not propose a new position encoding and used them as plug-ins. Furthermore, with a newer position encoding by Anonymous et al. 2022, on NC1, results are significantly better, however, on ZINC and MUTAG, values are less than SpecTRA + GCKN+ 3RW. Hence, the choice to use these encodings as plug-ins further justifies our claim that position encodings are yet not dataset agnostic to achieve a unanimous SotA.
> >
> > Kind regards,
> >
> > Authors of SpecTRA

---

> ### Author Response · Authors · 2021-12-02
> **Follow-up on Rebuttal from Authors**
>
> Dear Reviewer VYTd,
>
> We thank you again for taking the time to review this work. We put our best efforts to prepare the rebuttal to your questions. We would very much appreciate it if you could engage with us with your feedback on our rebuttal. We would be glad to answer any further questions and clarify any concerns. We also hope we clarified concerns regarding theorem and experiments as suggested by you.
>
> Also, if you are satisfied with our answers, please consider revising the score.
>
> With best regards,
>
> Authors of SpecTRA.

---

### Official Review · Reviewer_7T9R · 2021-11-03

**Correctness:** 2
**Technical Novelty And Significance:** 2
**Empirical Novelty And Significance:** 2
**Recommendation:** 3
**Confidence:** 5

**Main Review:**

+ The paper is well-structured.

+ The paper lacks its motivation. I can understand the motivation for filtering graphs for GNNs (Gao et al., 2021), but adapting this existing motivation to argue that the transformer cannot effectively segregate the noise from the signal is not enough. Some recent models (Mialon et al., 2021; Kreuzer et al., 2021) and SpecTRA are only suitable for small-size graphs due to utilizing the self-attention networks on the whole input graph. However, it is worth noting that these recent models ignore Graph-BERT [1] and Graph-Transformer [2].  Both Graph-BERT and Graph-Transformer can work quite well for node and graph classification and can be adapted for large-size graphs as they employ the self-attention networks on neighbors to take the local structures into account, which the recent models (Mialon et al., 2021; Kreuzer et al., 2021) and SpecTRA are missed.

+ Furthermore, it is straightforward to directly adapt Equation 6 for H^h with the learnable parameter \alpha instead of computing \alpha from the attention weights in Equation 9. Equation 9 is the main contribution in SpecTRA, but why does SpecTRA need it? What is the motivation behind it? What are the results of directly adapting Equation 6 with the learnable \alpha?

+ Regarding Equation 10, x^h_i is the vector at node x_i in the h-th attention head obtained from Equation 9. Is x^h_i actually from Equation 8? Because if it is from Equation 9, it means that SpecTRA constructs GCNs on top of the vanilla multi-head self-attention networks, followed by spectral filterings. This makes a confusion as if it is correct, SpecTRA is an extension of Graph-Transformer with additional using spectral filterings inspired from Gao et al., (2021).

+ SpecTRA lacks novelty as it is incremental. The performance gains are marginal.

+ The obtained results mainly come from the existing positional encoding schemes (Kreuzer et al., 2021). Without the existing positional encoding schemes, SpecTRA-Base is outperformed by other existing models such as GIN. Therefore, the main contribution shown in Equation 9 is incompetent.

+ The paper misses the results of the closely related works Graph-BERT, Graph-Transformer.

+ Why are the GIN and GCN results on MUTAG and NCI reported in Table 1 different from the original paper?

[1] Graph-Bert: Only Attention is Needed for Learning Graph Representations. https://github.com/jwzhanggy/Graph-Bert

[2] Universal Graph Transformer Self-Attention Networks. https://github.com/daiquocnguyen/Graph-Transformer


**Summary Of The Paper:**

This paper proposes SpecTRA to construct spectral filterings on top of vanilla multi-head self-attention networks to have coefficients for each input graph. Specifically, SpecTRA computes the filter coefficients from the attention weights of vanilla multi-head self-attention networks by defining a mapping function (i.e., Equation 9) from the attention space to the filter coefficient space.

**Summary Of The Review:**

The paper lacks the motivation and the proposed model SpecTRA is incremental.

---

> ### Author Response · Authors · 2021-11-14
> **Thank you and Author's response to reviewer's comments**
>
> We sincerely thank the reviewer for taking time out of their busy schedule and voluntarily spending time reviewing our work. Please find answers
>  - Question: not comparing with Graph-BERT/Graph-Transformer
>     - Answer: Work by Dwivedi et al. 2020 (GT model in our paper) made a strong case in their work regarding limitations of Graph-Transformer/Graph-BERT, hence, we and (may be SAN/Mialon et al) skip these works from comparison.  GT authors explain that Graph-BERT employs a combination of several positional encoding schemes to capture absolute node structural and relative node positional information. Hence, their main contribution is a position encoding into a vanilla transformer similar to GT, Mialon et al, and SAN. Empirical results by Dwivedi et al clearly suggest that Graph-BERT reports quite low values on large datasets such as PATTERN and CLUSTER - Table 3 in their work: https://arxiv.org/pdf/2012.09699.pdf. For Graph-Transformer, Dwivedi et al. further argued regarding limitations of this model by explaining that this model proposed a coordinate-based position encoding.
> - Question: Why is filtering needed and clear motivation
>     - Answer: There are several works such as {1,2} besides Gao et al in GNN that established the hypothesis that static low or high pass frequencies do not capture the full graph spectrum. We have a similar motivation that such issues also exist in transformers as after all transformers can be special cases of GNNs {3}. Hence, in addition to position encodings, the capability of filtering noise may also benefit. This is one reason SpecTRA shows improvement on vanilla transformers in Research question 1.
>
> - Question: straightforward to directly adapt Equation 6 :
>     - Answer: Directly adapting $\alpha_h$ as a learnable parameter is a SpecTRA-static configuration. This does not learn the filter components from the graph and so a static filter is being learned. The motivation of learning the filter coefficients dynamically from the graph is that if different components of the spectrum are “useful” for different graphs, then this will not be captured by the static (even though learnable) filters, keeping a limit on the parameters.  In table 3, SpecTRA-Base outperforms SpecTRA-static, thus supporting the hypothesis.
>
> - Question: Regarding Equation 10:
>     - Answer: In equation 10 the input graph is obtained from the attention heat map obtained from equation seven and the node attributes are initiated using all one vector of size  $R^k$. The input is not directly taken from the output in equation 8 of the transformer module. Our rationale behind using the attention map to learn the filter coefficients is that we wanted to learn the filters dynamically by observing their spatial connectivity. Thus, we hypothesize that the transformer can learn well. It is analogous to observing a one-dimensional signal, viewing the time intervals where the signals repeat (or are similar), and determining the frequency components. SpecTRA learns dynamic filters depending on the spatial connectivity of the graph, unlike Gao et al., which learns only in multiple subspaces but not graph-specific filters. As can be seen in the synthetic experiments in table 9 in appendix, the results using dynamic filters are better, justifying the applicability.
>
> - Question: Regarding SpecTRA being incremental
>     - Answer: None of the recent approaches on transformers change in the vanilla transformer except inducing external position encodings. Our school of thought is to fundamentally change how attention sub-spaces can be used to filter noise and provide interpretability. For NC1, GraphiT reports 81.4,  GIN with 81.7, and SAN with 80.5. We report 83.7 which is in the range of previous baselines' jump on its predecessor model (similarly on MUTAG and others). Yes, on PATTERN, we report similar values as SAN, however, additionally providing interpretability and the ability to filter the noise by modifying vanilla transformer.
>
> - Question: Regarding jump of position encoding:
>     - Answer: It is an expected behavior. SpecTRA shows improvement over the vanilla model: research question 1. Both models induced with recent position encoding schemes show improvement because neither transformer nor SpecTRA-Base has capability to capture graph topologies. We presented experiments with position encodings to show that the filtering module plus position encoding complements each other, and there is a jump in comparison to Mialon et al or SAN when their vanilla transformer is replaced by SpecTRA-Base.
>
> We get values and experiment settings of MUTAG and NC1 from Mialon et al. Hence, the values differ from original work.
>
> {1} ICLR2021 https://arxiv.org/abs/2006.07988.
>
> {2}  AAAI 2021: https://arxiv.org/abs/2101.00797
> {3} Joshi, Chaitanya. "Transformers are graph neural networks." The Gradient (2020).
>
> We sincerely hope our data points clarify the reviewer's concern. If yes, would you be kind to us to increase our score?

---

> ### Author Response · Authors · 2021-12-02
> **Follow-up from Authors on Rebuttal**
>
> Dear Reviewer 7T9R,
>
> We thank you again for taking the time to review this work. We put our best efforts to prepare the rebuttal to your questions. We would very much appreciate it if you could engage with us with your feedback on our rebuttal. We would be glad to answer any further questions and clarify any concerns. We also hope we clarified concerns regarding Graph-BERT, and Graph-Transformer by pointing towards evaluation done by Dwivedi et al.
>
> Also, if you are satisfied with our answers, please consider revising the score.
>
> With best regards,
>
> Authors of SpecTRA.

---

### Official Review · Reviewer_5mRd · 2021-11-03

**Correctness:** 4
**Technical Novelty And Significance:** 3
**Empirical Novelty And Significance:** 3
**Recommendation:** 5
**Confidence:** 2

**Details Of Ethics Concerns:**

None.

**Main Review:**

Strength:
1. This paper has clear motivation and goal; that is since the transformer model naturally gives diverse attention sub-spaces, which correspond to the multiple filters covering the spectrum of the graph. Under these settings, the goal of the model is then to learn the filter coefficients; this process also naturally provides interpretability.
2. This paper has solid theoretical proof.
3. The experiment is pretty comprehensive; it includes many models from recent years as baselines and the comparison is done on six different datasets under three different tasks.
4. Overall this paper is well organized where the authors clearly know what they are talking about.

Weakness:
1. From the experiment section we can see that the proposed model can do pretty well on some tasks while doing worse on other datasets. This variance in performance could suggest that the proposed model need certain prerequisite to be fully functional. More datasets are needed to fully test the functionality of this model.
2. The proposed model is usually combined with other modules to achieve the best performance while the base version of it is definitely not the best in comparison to other models. More ablation studies might be necessary to justify the role that the proposed model plays in the system.

**Summary Of The Paper:**

This paper proposes a novel model called SpecTRA to optimize the performance of the transformer model on the graphs. This proposed model induces a spectral module into the original transformer architecture to enable it to decomposite graph spectrum and filter the frequency domain and thus leading to more useful information.

**Summary Of The Review:**

This paper proposes a model that tries to improve the performance on the transformer on graph structure data. The proposed model has pretty good motivation and reasonable theory support, with the performance being relatively good in comparison to other baselines on some tasks. Overall, this is a solid paper, but it might more experiments to justify the importance of the proposed model.

---

> ### Author Response · Authors · 2021-11-14
> **Thank you and Author's response to reviewer's comment**
>
> We sincerely thank the reviewer for taking time out of your busy schedule and voluntarily spending time reviewing our work. We provide a step-by-step explanation for points raised by the reviewer. **Also, as suggested by you, we provided an additional experiment using under-submission position encoding in ICLR2022 (please see common comments to all reviewers as above), illustrating the efficacy of the proposed approach**.
> - Question: SpecTRA needs pre-requisite:
>   - Answer: In the rebuttal version of our paper, we have divided Table 1 into two parts: only comparison with vanilla transformer as it has been our main focus, and second is the impact of position encoding. SpecTRA is a variant of a vanilla transformer. Neither of these models have an explicit dependency on position encodings to be fully functional. Hence, to be fully functional, none of these models explicitly needs extra modules. Hence, if we just compare SpecTRA-Base with a vanilla transformer, performance increases quite well (RQ1 in our paper).
>
> - Question: Regarding the model and its variated performance across tasks:
>   - Answer: It's a valid observation. Let us now take a step back and just look at recent transformer-based Models in Table2 of the rebuttal version of our paper.  Mialon et al., 2021 that use vanilla transformers report the best results on NC1 and MUTAG and report fewer values on other datasets such as PATTERN and CLUSTER. Similarly, SAN, which also uses vanilla transformers, reports fewer values than Mialon et al on MUTAG and NC1. Mialon et al/SAN is a vanilla transformer+their proposed position encoding as contributions. Now:
>     - When we replace SAN/Mialon et al's vanilla transformer with SpecTRA-BASE, there is a positive impact on average. For example, SpecTRA variants such as SpecTRA+GCKN+3RW exchange vanilla transformers with SpecTRA-Base in Mialon et al. Compared to original values reported by authors on NC1, MUTAG, ZINC, our values increase. Hence, on nearly the same parameters, the jump is coming from changes made by us in the vanilla transformer which is the ability to filter out noise from the signals and empower transformers with this capability. Hence illustrating the clear effectiveness of our proposed dynamic filtering module.
>     - Similarly on PATTERN, CLUSTER, MOLHIV, with position encoding of SAN replacing their vanilla model with SpecTRA-Base, values are either comparable (pattern), or increased (cluster, MOLHIV). Just to make a statement that position encodings are dataset specific, we ran 8 SpecTRA configurations, on 6 datasets to run 48 experiments plus running baselines on datasets where their performance is limited. We hope such extensive evaluation to conclusively make a point makes sense.
>     - Hence loosely speaking, the dynamic filtering module complements the performance jump of position encoding. However, the vanilla transformer needs both- our changes plus position encoding to compete with the best models. Transformers for graphs is a newer domain and a long way to go for the research community. Ours is the first attempt to propose a completely new direction within the transformer on graphs domain to bring this dynamic filtering angle, besides widely studied position encoding school of thought.
>     -  The variation in the performance is largely due to the limitations of the existing position encoding schemes on different datasets. This behavior is similarly observed for Vanilla transformer as well as SpecTRA-Base.
>  - Question: Regarding more datasets to test model functionality
>     - Answer: Apart from the four datasets used by SAN, we added two popular datasets MUTAG and NC1. Furthermore, we also did an extensive evaluation on three different synthetic datasets to evaluate SpecTRA-Base capability in filtering noise in Table 9 of Appendix A.4. We believe such extensive evaluation enabled us to make conclusive remarks on SpecTRA's capabilities.
>  - Question: Regarding additional Ablations:
>     - Answer: To understand the effect of dynamic filtering, we added ablations concerning a static filter (Table 3) that showed a limited performance. Also to illustrate that SpecTRA is not tied to the Chebyshev filter, we provided ablation with ARMA filter in Table 3. Due to the limitations of 9 pages and keeping the focus of the paper intact, in Table 9 (rebuttal version) of Appendix A.4,  we studied the effect of different filter orders. We hope such systematic ablations to support where the performance gain is coming in SpecTRA-Base are sufficient due to the scope of the work.
>
> We sincerely hope our data points clarify the reviewer's concern. If yes, would you be kind to us to increase our score?

---

> ### Author Response · Authors · 2021-12-02
> **Follow-up on Rebuttal from Authors**
>
> Dear Reviewer 5mRd,
>
> We thank you again for taking the time to review this work. We put our best efforts to prepare the rebuttal to your questions. We would very much appreciate it if you could engage with us with your feedback on our rebuttal. We would be glad to answer any further questions and clarify any concerns. Also, in common comments to all reviewers, please find extended experiments as suggested by you.
>
> Also, if you are satisfied with our answers, please consider revising the score.
>
> With best regards,
>
> Authors of SpecTRA.

---

### Official Review · Reviewer_iKKm · 2021-11-06

**Correctness:** 3
**Technical Novelty And Significance:** 3
**Empirical Novelty And Significance:** 2
**Recommendation:** 5
**Confidence:** 4

**Main Review:**

The paper discusses augmenting the Transformer structure to better suit the nature of graph data modeling. My main concern is that the paper itself does not compose a coherent story of why the GNN layer should be effective and makes a difference as a whole.

## Pros
The paper provides a good way to interleave GNN and Attention layers. Compared to simply injecting graph structures into positional embeddings, this method gives a novel way to fuse benefits from both sides.

## Cons
* The descriptions and notations by the paper are preventing a coherent understanding. e.g. the superscript $h$ (e.g. $\alpha^h$) is used to represent the corresponding head number while superscript $l$ is used to represent the layer number. When combined with the output representation $x$, it would be unclear for a first-time reader to notice the meanings of $x^h$ and $x^l$ are different.
* The method itself is hard to be in the SOTA class. For example, in Table 1, even for SpecTRA+GCKN+3RW which is the best among 3/6 datasets, it can still not outperform the previous SAN-Sparse one on the remaining 3/6 datasets. Given its added complexity compared to SAN (they share most of the parts in the positional embedding scheme), it’s hard to justify its effectiveness.
* A fair Transformer comparison is needed. It would be beneficial to state the added number of parameters compared to the baseline Transformer. As in most cases that would induce larger model capacity.
* Minor:
  * The current draft itself can be more polished to be a qualified submission. There are some typos and unclear statements:
  * Equation (6) double `)`
  * Equation (8) Missing $V$
  * 2 lines above Equation (11)$h^{th}$ -> $h$-th
  * Figure 3, it would be better to show unsmoothed filter frequencies for limited-point graphs (e.g. (a)).


**Summary Of The Paper:**

This paper proposes a framework that adds in an extra GNN layer after the Attention layer of Transformer, which enables better quality modeling in graph data. In particular, the GNN layer learns a per-head transformation that imposes learnable filter coefficients, which demonstrates better performance than static filters coefficients.


**Summary Of The Review:**

The proposed method gives a new way of enhancing Transformer models in the graph domain. However, the experiments are not sufficient to justify the efficacy of the new model given its added complexity.

---

> ### Author Response · Authors · 2021-11-14
> **thank you and Author's response to the reviewer's comments**
>
> We sincerely thank the reviewer for taking time out of busy schedule and voluntarily spending time reviewing our work. We also modified paper to incorporate reviewers suggestions to clearly present our contribution.:
> - Question: Regarding notations:
>   - Answer: many thanks again for catching formatting inconsistency wrt symbols. We have submitted the revised version with modification of issues in the formatting, and further polished section Introduction, Abstract, and section 4.1 based on other reviewer's comments.
>
> - Question: Regarding unsmoothed figure:
>   - Answer: We have modified figure 3a to show unsmoothed filter frequencies in the rebuttal version.
>
> - Question: Regarding SpecTRA being in SoTA Class:
>   -  Answer: We have divided Table 1 into two parts: only comparison with vanilla transformer as it has been our main focus where our model performs quite well against a vanilla transformer, and second is the impact of position encoding in Table2.
> And yours is a very valid point. The Vanilla Transformer shows limitations for Graphs, which is a well-known fact. But Why? One school of thought to increase its performance is to induce position encoding, which SAN, Mialon et al., and Graphformer do. SAN is SoTA for 2/6 datasets (Pattern, Cluster), whereas Mialon et al 2021 is a SotA on 2/6 datasets (MUTAG and NC1), and Graphformer is on ZINC and MOLHIV. Hence in Table 2 (rebuttal version), if we just compare recent transformer-based models assuming there is no SpecTRA, there exists no clear definition or class of “which transformer model is a clear SotA across tasks, across datasets”? Hence, position encoding schemes don't seem to work as robust solutions across datasets and tasks. This prompted us to study if transformers can be empowered to filter out noise from graph signals.
>   - The empirical comparison (our research question one) clearly illustrates that against vanilla transformers, SpecTRA is much better on all tasks agnostic of dataset. However, when we induce position encoding, behaviour is exactly what is expected: with position encoding schemes of Mialon et al (basically replacing vanilla transformer in Mialon et al with SpecTRA-Base), SpecTRA gets SotA for 3/6 datasets because we improve vanilla transformer capabilities (SpecTRA+GCKN+3RW ). On other dataset this position encoding finds limitation with vanilla transformer, that limitation continued with SpecTRA-base too.  Similarly with SAN’s position encodings, achieving comparable (Pattern) or better (on Cluster/MOLHIV) where original SAN performs well. It signifies two things:
>     1) Proposed position encoding schemes do not work agnostic of tasks neither for a vanilla transformer nor for its variant such as SpecTRA. We as a research community have not yet achieved a universal position encoding that is enough for transformer making it work for graph.
>     2) SpecTRA, when improving capabilities of the vanilla transformer model, compliments performance jump achieved from position encoding schemes, depending on their original SotA datasets where the vanilla transformer was used as a base. Hence, it's not just position encoding, but its combination with filtering capabilities performs well.
>
> - Question: Regarding complexity:
>    - Answer: For SpecTRA configurations with position encodings, our parameters are nearly the same compared to SAN or Mialon et al’s proposed position encoding as provided in the appendix section of our work. Hence, whatever performance jump we see after inducing position encoding is due to SpecTRA-Base's ability to filter out the noise and capture a complete graph spectrum. In SpecTRA's variants with position encoding, only the base transformer changes. The rest all remain the same as the original implementation of SAN/Mialon et al. Based on your comments for a clear comparison, we have divided table 4 of the previous version in the newer submitted version with tables 5,6,7 in the appendix to provide a fair comparison of the model size against each model configuration. We hope it answers your concern regarding our model being more complex/heavy in parameters than SAN.
>
> - Question: Vanilla transformer Vs SpecTRA parameters:
>   - Answer: We added a detailed parameter comparison against the vanilla transformer in the appendix section in the rebuttal version. Also, please find them in Appendix table 5.
>
> We sincerely hope our data points clarify the reviewer's concern. If yes, would you be kind to us to increase our score?

---

> ### Author Response · Authors · 2021-12-02
> **Follow-up on Rebuttal from Authors**
>
> Dear Reviewer iKKm,
>
> We thank you again for taking the time to review this work. We put our best efforts to prepare the rebuttal to your questions. We would very much appreciate it if you could engage with us with your feedback on our rebuttal. We would be glad to answer any further questions and clarify any concerns. We also provided extended experiments to support our claims and your concerns regarding position encoding impact
>
> Also, if you are satisfied with our answers, please consider revising the score.
>
> With best regards,
>
> Authors of SpecTRA.

---

### Author Response · Authors · 2021-11-22
**Extended Evaluation (significant improvement on NC1) to further Strengthen proposed approach- Authors Comment**

Dear Reviewers,

Many thanks again for spending time on our work. We would like to bring your attention to our argument in the paper and also in rebuttal that position encodings are not the only solution for transformers on Graphs. In the latest proposed position encoding (under submission in ICLR2022, Anonymous et al 2022: https://openreview.net/pdf?id=wTTjnvGphYj), we replaced their transformer with SpecTRA-Base to further support our claims in the paper. However, due to the limited time of the discussion period, we could only get results of three datasets, which we post below:

| Model   Name  | MUTAG    | NC1     | ZINC  |
| :------------- | :----------: | -----------: | -----------: |
| SpecTRA + GCKN+ 3RW (reported by us in the paper) | **92.9 ± 1.6**  |83.0 ± 0.5  |**0.068 ± 0.002**  |
| LSPE+Transformer (Anonymous et al. 2022) | 88.9± 2.8  |83.5± 1.8  |0.104±0.004   |
| LSPE+ SpecTRA-Base (ours) | 89.0± 1.7 | **86.1± 2.2**|  0.098±0.002  | \| |

The above values signify the following:
  - The newly proposed position encoding behavior in a transformer further strengthens our arguments that these encodings are not dataset agnostic to impact transformer performance to become unanimous SoTA across datasets.
  - When we replace the transformer with SpecTRA-Base in their implementation, we achieve a jump in the performance. In fact on NC1, LSPE+ SpecTRA-Base performs even better than SpecTRA + GCKN+ 3RW (ours) with 83.0 ± 0.5 values. Hence, SpecTRA with the newly proposed position encoding LSPE is now state-of-the-art on NC1 by a significant margin.
  - On ZINC and MUTAG, still, the previously proposed position encoding by Mialon et al (SpecTRA + GCKN+ 3RW) complements SpecTRA's ability the best to remain SotA.
  - Hence, our work shifts the paradigm of the current popular school of thought to improve vanilla transformers through position encodings. SpecTRA-Base's ability to dynamically choose filters to remove noise in graph signals complements the performance of position encoding established the necessity of our work in this domain.

We sincerely hope, our additional experiments with newly proposed position encoding (Anonymous et al 2022) also strengthened our rationale of not proposing a new position encoding, and support evaluation choices to use position encodings as plug-ins.  It also justifies why single position encoding induced in SpecTRA does not achieve a unanimous state-of-the-art across datasets.

thank you,

Authors of SpecTRA

---

### Decision · Program_Chairs · 2022-01-20

**Decision:**

Reject

**Comment:**

This paper proposes a novel approach to include graph information into Transformers. Reviewers expressed concerns on 2 main issues -

1) The exact architecture proposed in the paper is not well motivated. In words of one of the reviewers 'I still do not understand why the authors learn the spectral GCN filter weights from the attention matrix of the transformer, which can have a completely different sparsity pattern than the input graph, instead of learning the filter weights from the graph itself, e.g., by using a GNN. '. Authors tried to provide an explanation in the response however I think it needs to be made much more rigorous for it to be well motivated.

2) The interplay between existing position encoding schemes and the proposed method. This point also confused couple of reviewers as the empirical results seem to be strongly influenced by the choices of position encoding. Authors, I think did a great job in addressing this concern by providing additional results during the response period.

Given the weak experimental results and lack of clear motivations I think the paper is not currently ready for acceptance.